# Small AntiMicrobial Peptide with In Vivo Activity Against Sepsis

**DOI:** 10.3390/molecules24091702

**Published:** 2019-05-01

**Authors:** Héloise Boullet, Fayçal Bentot, Arnaud Hequet, Carine Ganem-Elbaz, Chérine Bechara, Emeline Pacreau, Pierre Launay, Sandrine Sagan, Claude Jolivalt, Claire Lacombe, Roba Moumné, Philippe Karoyan

**Affiliations:** 1Sorbonne Université, École Normale Supérieure, PSL University, CNRS, Laboratoire des Biomolécules, LBM, 75005 Paris, France; hel.boullet@gmail.com (H.B.); faycal.bentot@gmail.com (F.B.); cherine.bechara@umontpellier.fr (C.B.); sandrine.sagan@sorbonne-universite.fr (S.S.); claire.lacombe.s@gmail.com (C.L.); roba.moumne@sorbonne-universite.fr (R.M.); 2Laboratoire Charles Friedel, UMR7223, École Nationale Supérieure de Chimie de Paris, 11 rue Pierre et Marie Curie, 75005 Paris, France; arhequet@gmail.com (A.H.); carine.ganem-elbaz@curie.fr (C.G.-E.); claude.jolivalt@sorbonne-universite.fr (C.J.); 3Inserm U1149, Labex Inflammex, Bichat Medical School, 75005 Paris, France; emeline.pacreau@inserm.fr (E.P.); pierre.launay@inserm.fr (P.L.); 4Faculté des Sciences et Technologie, Univ Paris Est-Créteil Val de Marne, 94000 Créteil, France; 5Kayvisa, AG, Industriestrasse, 44, 6300 Zug, Switzerland; 6Kaybiotix, GmbH, Zugerstrasse 32, 6340 Baar, Switzerland

**Keywords:** polycationic β-amino acids, small antimicrobial peptides, sepsis

## Abstract

Antimicrobial peptides (AMPs) are considered as potential therapeutic sources of future antibiotics because of their broad-spectrum activities and alternative mechanisms of action compared to conventional antibiotics. Although AMPs present considerable advantages over conventional antibiotics, their clinical and commercial development still have some limitations, because of their potential toxicity, susceptibility to proteases, and high cost of production. To overcome these drawbacks, the use of peptides mimics is anticipated to avoid the proteolysis, while the identification of minimalist peptide sequences retaining antimicrobial activities could bring a solution for the cost issue. We describe here new polycationic β-amino acids combining these two properties, that we used to design small dipeptides that appeared to be active against Gram-positive and Gram-negative bacteria, selective against prokaryotic versus mammalian cells, and highly stable in human plasma. Moreover, the in vivo data activity obtained in septic mice reveals that the bacterial killing effect allows the control of the infection and increases the survival rate of cecal ligature and puncture (CLP)-treated mice.

## 1. Introduction

If the discovery of antibiotics is one of the major medical breakthroughs of the last century, bacterial resistance has consecutively emerged as a main medical problem [1]. Indeed, the number of infections caused by bacterial strains resistant to conventional antibiotics is rising and despite the success of genomics in identifying new essential bacterial genes, there is a lack of sustainable leads in antibacterial drug discovery to address these increasing multidrug-resistant (MDR) microorganisms [2]. The search for novel antibiotics with original mechanism of action is of particular interest. In this context, Antimicrobial Peptides (AMPs) are considered as an inspirational source for future antibiotics [3,4]. Indeed, although their mechanism of action is still a matter of basic research, it is generally admitted that most of them act directly on the bacterial membrane (membranolytic) and thus likely escape the mechanisms of bacterial resistance [5]. Although AMPs present considerable advantages as new generation antibiotics, their development as therapeutics is still limited by peptide drawbacks, such as their potential toxicity, susceptibility to proteases, and high manufacturing costs. To overcome these limitations, different strategies have been investigated: The use of unnatural amino acids is anticipated to enhance their proteolytic stability [6], while the identification of small antimicrobial peptides (SAMP) [7] with sequence length ranging from 2 to 10 amino acids is suggested as an interesting solution for the cost issue. Small non peptidic scaffolds that mimic their mechanism of action have also been recently reported [8,9].

AMPs are usually amphipathic sequences and contain several basic residues, i.e., lysine and arginine, as well as a hydrophobic core, which are critical for their activity. The lysine and arginine side-chains are positively charged at physiological pH and direct these amphiphilic peptides to the anionic surface of bacterial cell membranes, allowing the interaction of hydrophobic residues with the hydrocarbon core of the lipid bilayer. In the aim of identifying minimalist sequence that act like AMP, the use of building blocks bearing multi-cationic groups at physiological pH could be an interesting strategy. Aussedat et al. have previously reported a small achiral tetravalent template, the “α-bis-arginine”, which contains twice the side chain of arginine, and thus increases the charge density of the peptide sequence [10]. Although a promising tool, the steric hindrance of the α-bis-arginine quaternary center adjacent to the amine and acid functions rendered its peptidic coupling difficult in SPPS or LPPS (Solid and Liquid Phase Peptide Syntheses). The use of additional non-bulky spacers such as glycine or β-alanine residues was necessary to incorporate this α-amino acid into peptides. Consequently, even if the number of charged residues could be reduced through the use of this multi-charged amino acid, the overall size of the peptide cannot be shortened. We report here new residues that combine the advantage of the α-bis-arginine but can be easily oligomerized leading to small peptides with potential therapeutic applications: the β^2,2^- and β^3,3^-*homo*-bis-arginine derivatives, homologated respectively on the carboxylate or on the amino side (Figure 1). We postulated that the additional methylene group of β-amino acids (in green in Figure 1) would limit the steric hindrance around the quaternary center (in red in Figure 1) and facilitate their incorporation into peptides. Oligomers of β-amino acids represent one of the most studied class of foldamers. Since the pioneer work of Seebach et al. [11], only few studies dealing with β^2,2^- or β^3,3^-amino acids have been reported in the literature [12,13,14]. Noticeably, while the use of lipophilic β^2,2^-amino acids has proven valuable for the design of both antibacterial [15] and anticancer peptides [16,17], geminally disubstituted residues with basic side-chains have not been reported so far.

We report here the syntheses of β^2,2^- and β^3,3^-bis-*homo*-ornithine/arginine, and their use to design small cationic peptides. These peptides were evaluated as antimicrobial agents against Gram-positive and Gram-negative bacteria, and their cytotoxicity against eukaryotic cells as well as their stability in human serum were assessed. This work led to the selection of a dipeptide as a lead for in vivo studies for the treatment of sepsis in mice. Remarkably, the in vivo results revealed that the bacterial killing effect of this cationic dipeptide allows the control of the infection and sustains the immune response in the remediation of sepsis.

## 2. Results

### 2.1. Amino Acids Syntheses

The β^2,2^- (**1** and **2**) and β^3,3^-bis-*homo*-ornithine derivatives (**3**) required for the synthesis of the cationic dipeptides were prepared suitably protected for dipeptide syntheses (Figure 2).

The β^2,2^-*homo*-bis-ornithine methyl ester **1** and the Fmoc-protected β^2,2^-*homo*-bis-ornithine **2** were both obtained from methyl cyanoacetate, respectively, in three and four steps (Scheme 1).

The double Michael addition on acrylonitrile [18] followed by selective reduction of the nitrile groups in γ-position over PtO_2_ and simultaneous Boc-protection of the resulting amines gave the key intermediate **4** with moderate yields (21%). Improvement of this yield could be realized using a large excess of Raney Nickel (50% Yields) but was not relevant for safety reason and large-scale synthesis. Reduction of the α-nitrile by Raney nickel catalyzed hydrogenation in methanol led to the amine-free, acid-protected β^2,2^-*homo*-bis-ornithine derivative **1** that could be directly used in peptide coupling on the amine side. The *N*-protected, acid-free counterpart **2** was obtained when the reduction of **4** was performed in the presence of sodium hydroxide, followed by a Fmoc-protection. Boc-protected β^3,3^-*homo*-bis-ornithine derivative **3** was obtained starting from *tert*-butyl benzyl malonate (Scheme 2).

The double Michael addition on acrylonitrile followed by selective benzyl ester hydrogenolysis using ammonium formate on palladium charcoal gave compound **5**. Arndt–Eistert homologation catalyzed by silver oxide led to compound **6** with 21% yields over the three steps. After deprotection of the *t*-Bu ester, the acid group was converted to Boc-protected amine via a Curtius rearrangement. Reduction of the nitrile groups in γ-position was then achieved by platinum oxide catalyzed hydrogenation. Finally, protection of the amino groups as Boc-carbamate and saponification of the methyl ester gave access to compound **3** readily usable for peptide coupling on the acid side. 

### 2.2. Peptides Design and Syntheses

With these compounds in hand, we have designed antimicrobial dipeptides inspired by the work of Svendsen and co-workers, who defined the minimal set of functional motifs required to develop short AMPs as two cationic charges and two bulky hydrophobic aromatic units [19,20]. Based on this minimalist pharmacophore model, they indeed developed promising antibacterial tripeptides composed of a central 2,5,7-tri-tertbutyltryptophan (Tbt) flanked by two arginine residues. These peptides have anti-infectious properties and have reached phase-II clinical studies [21,22,23]. Several other groups have then reported the successful implementation of this pharmacophore model [24,25,26]. Starting from the peptide reported by Svendsen et al., the two arginine residues were replaced by one dicationic amino acid, leading to dipeptides **8**–**13** containing a tryptophan derivative (Trp or Tbt) and a dicationic β^2,2^- or β^3,3^-amino acid: Trp-β^2,2^-*h*-bis-Orn-OMe (**8)**, Tbt-β^2,2^-*h*-bis-Orn-OMe (**9**), Gdm-Trp-β^2,2^-*h*-bis-Arg-OMe (**10**), Tbt-β^2,2^-*h*-bis-Arg-OMe (**11**), Gdm-Tbt-β^2,2^-*h*-bis-Arg-OMe (**12**), and β^3,3^-*h*-bis-Arg-Tbt-OMe (**13**) (Figure 3). In order to investigate the effect of the positive charge segregation on the antimicrobial activity of the compound [27], we also synthesized peptide **14** (Gdm-β^2,2^-*h*-bis-Arg-Tbt-OMe), in which the sequence of dipeptide **11** is reversed.

To evaluate the ease of coupling of these new beta derivatives against their alpha counterparts, both liquid and solid phase peptide syntheses were tested. Compounds **8**–**12** were prepared by coupling the corresponding tryptophan derivatives (Boc-Trp-OH or Fmoc-Tbt-OH) with the β^2,2^-*h*-bis-ornithine methyl ester **1** in solution, using HBTU as a coupling agent, in the presence of DIEA, in DMF (Scheme 3).

The fully protected dipeptides **15** and **16** were obtained from Boc-Trp-OH and Fmoc-Tbt-OH, in respectively 99% and 70% yields. Noticeably, the α-bis-ornithine derivative coupling failed in the same conditions. Deprotection of the amines gave access to the corresponding β^2,2^-*h*-bis-Ornitine derivatives **8** and **9**. Introduction of the guanidinium group (Gdm) on these two compounds followed by Boc-deprotection using a TFA cocktail led to the β^2,2^-*h*-bis-Arg derivatives. While a unique tri-guanylated compound was obtained for the tryptophan containing dipeptide **10**, two products were isolated for the Tbt-derived compound in respectively 59% and 22% yields: One with the guanidinium groups on the side chains of the amino acids (**11**) only, and one with an additional guanidinium group on the β-amine (**12**). 

The synthesis of peptides **13** and **14** was achieved by SPPS, starting from a HMBA resin-bound Tbt (Scheme 4).

In both cases coupling of Fmoc-β^2,2^-*h*-bis-Orn-OH **2** and Boc-β^3,3^-*h*-bis-Orn-OH **3** was achieved through HATU activation, in the presence of DIEA, in DMF. However, because of the steric hindrance of its carboxyl group, heating at 50 °C as well as a second coupling round were necessary to ensure the complete conversion of **2**. As anticipated, the improved reactivity of the carboxyl group of this residue with its β^3,3^-counterpart confirms that an additional methylene near the quaternary center is an effective strategy to facilitate the incorporation of the bis-ornithine derivative into a peptide sequence. After piperidine-mediated Fmoc-deprotection and/or removal of the acid labile protective groups by treatment with a trifluoroacetic acid (TFA)-triisopropylsilane (TIS)-H_2_O cocktail, introduction of the guanidine moiety was performed using an excess of 1,3-di-Boc-2-(trifluoromethylsulfonyl)guanidine in DMF, in the presence of triethylamine, followed by removal of the Boc-protective groups. Cleavage of peptides **13** and **14** from the resin was achieved by treatment with methanol in the presence of DIEA and DMF giving direct access to the methyl ester protected dipeptide. Compound **14** was obtained as a tri-guanylated derivative. On the contrary, as expected, the steric hindrance of the quaternary β-amino group of compound **13** prevents any reaction on the backbone amine. In addition, NMR analysis confirmed that peptide **13** was only guanylated on the amine side-chains. Several studies have reported that the N-terminal capping of cationic peptides with a fatty acid moiety enhances their antimicrobial activity [28,29]. Thus, in order to further improve the potency of **11**, an additional hydrophobic group was incorporated, first on the N-terminal end of the sequence. (Figure 4, peptides **17**–**19**).

We also evaluated whether such a capping effect could be also observed in this series of peptides. The biological activities of Fmoc-protected derivatives Fmoc-Tbt-β^2,2^-*h*-bis-Orn-OMe **17a** and Fmoc-Tbt-β^2,2^-*h*-bis-Arg-OMe **17b** were synthesized in addition to the ones of the two compounds **18** and **19** capped through a more robust amide bond at their N-terminal end. All peptides were purified to >95% homogeneity by preparative RP-HPLC and the mass of each purified peptide was checked by MALDI MS (see Appendix A). 

Finally, in order to study the influence Trp- and Tbt derivatives, we compared the retention time in RP-HPLC of selected peptides (Figure 5).

### 2.3. Biological Activities

#### 2.3.1. Antimicrobial, Hemolytic, and Cytotoxic Activities and Serum Stability

The antibacterial activities of the peptides were then investigated in the conditions reported by Svendsen, by determining the Minimal Inhibitory Concentration (MIC, µg/mL) on six strains of bacteria; three Gram-positive, *Staphylococcus aureus* ATCC25923, *Enterococcus faecalis* ATCC29212, and the methicillin resistant *Staphylococcus aureus* SA-1199B, and three Gram-negative, *Escherichia coli* ATCC25922, *Pseudomonas aeruginosa* ATCC27853, and *Acinetobacter baumannii* ATCC19606 [14] (Table 1). The tri-peptide Arg-Tbt-Arg-NH_2_ reported by Svendsen (called here peptide A), and the dipeptide Tbt-Arg-OMe (called here peptide B), were used as positive controls of our experimental conditions.

The hemolytic and cytotoxic activities against human cells of all active peptides were assessed (Table 1, Figure 6 and Figure 7 and Appendix A).

#### 2.3.2. Interaction with Membrane Model

Although the mechanism of action of AMPs is still an active field of research, it is generally admitted that a common primary mode of action involves the disruption of cellular membrane. In order to get some insights into the mechanism of action, biophysical studies were conducted with membrane model. We used the intrinsic fluorescent properties of the tryptophan residue, as initial analysis of the bactericidal mechanism [30]. Depending on its environment in peptides, the wavelength of the fluorescence light emitted by the aromatic tryptophan residues varies. In a polar environment (water), λmax is circa 357 nm, whereas in a non-polar one, λmax shifts to shorter wavelengths (blue-shift). Moreover, the emission intensity increases when the tryptophan residue enters into a hydrophobic environment [31]. We therefore recorded the fluorescence of the most active peptide **11** and compared it to the inactive one **10** (Figure 8).

#### 2.3.3. In Vivo Experiment Studies

In vivo experiment studies were conducted on septic mice. Sepsis is a life-threatening condition described as a syndrome of infection complicated by acute organ dysfunction. It is still a leading cause of death in intensive care units despite early antibiotic strategies to control bacterial infection [32]. Therefore, the rapidity and efficacy of antibacterial strategies are highly connected to the outcome of this acute disease and patient survival. After acute cecal ligature and puncture (CLP), peptide **11** or PBS (negative control) were injected to mice and survival was observed (Figure 9).

## 3. Discussion

We have designed small AMPs based on new polycationic β-amino acids, β^2,2^- and β^3,3^-*homo*-bis-ornithine derivatives. These moieties mimic the cationic side chains of two lysine residues or two arginine residues and thus allow shortening the cationic AMP size. Their combination with the supertryptophan residue (2,5,7-tri-tertbutyltryptophane) reported by Svendsen and co-workers allows obtaining highly active antimicrobial dipeptides. They exhibit activity in the range of 2 to 16 µg/mL (Table 1), values that are promising for compounds to enter into clinical trials. Among the different peptides tested, several β^2,2^- and β^3,3^-*bis* cationic derivatives (peptides **11**–**14**) were potent killing agents against the different strains, with MIC values comparable to or lower than that of the positive controls, and no significant difference was observed between the compound derived from the β^2,2^- (**11**) and β^3,3^-*h*-bis-Arg (**13**). Noticeably, the β^2,2^-amino acid derivatives are easier to synthesize. 

Some structure activity relationships can be drawn from these results. First, the importance of the guanidinium groups for the antimicrobial activity is highlighted, since peptide **9**, containing the β^2,2^-*h*-bis-Orn, shows little antimicrobial activity against all strains (except *S. aureus*) compared to the β^2,2^-*h*-bis-Arg analog **11**. This net difference in the antimicrobial activity of arginine- and lysine-containing compounds agrees with the literature and is believed to result from the stronger ability of the guanidinium group to form bidentate hydrogen bonds with the phosphate moiety of phospholipid polar heads, in addition to electrostatic interactions [33]. Oppositely, the absence of difference in the antimicrobial activity of peptides **11** and **12** indicates that the additional guanidinium group on the β-amine has little influence, suggesting that the cationic group on the N-terminal end is not involved in the pharmacophore of the peptide.

Another important point is the positive influence of the *t*-Bu group on the tryptophan moiety, similar to the peptide reported by Svendsen et al. Indeed, in comparison to **11** or **12**, peptide **10** presents no activity on the tested strains. This lack of activity can be related to the lower lipophilicity of tryptophan compared to the Tbt derivatives **11** and **12**, confirmed by its lower retention time in RP-HPLC (Figure 5) together with its lower capacity to interact with membrane. Indeed, the larger size of Tbt compared to Trp (around 2.5-fold) could allow a deeper penetration of this hydrophobic residue into the phospholipid bilayer and an effective disruption of the membrane that is not allowed by the smaller indole moiety. In order to evaluate this hypothesis, we recorded the fluorescence of the active peptide **11**, and compared it to the inactive peptide **10**. The tryptophan fluorescence spectra of both peptides in aqueous buffer had a maximum emission at 355 nm. Addition of increasing concentration of large unilamellar vesicles (LUVs), prepared from phospholipids directly extracted from *S. epidermidis*, showed large blue-shift (near 35 nm) in the emission maxima of peptide **11**, characteristic of the embedding of Trp side chain into the hydrophobic medium of the negatively charged phospholipid (Figure 2). For peptide **10**, the blue-shift was 10 nm smaller with apparent binding constant K_L_ (lipid concentration that induced 50% of maximal blue-shift) about 3 times lower for peptide **11** (90 ± 2 mg·mL^−1^ s and 100 ± 1.5 mg·mL^−1^*,* respectively with *S. aureus* LUVs and *E. coli* LUVs) than for peptide **10** (280 ± 1.5 mg·mL^−1^ and 500 ± 6 mg·mL^−1^ with *S. aureus* LUVs and *E. coli* LUVs). These preliminary biophysical studies on the interaction of **11** with model membrane suggested that this compound indeed could act as an antimicrobial peptide, by destabilizing the bacterial membrane. We are aware that deeper investigations might be performed in order to assess the mechanism by which this membrane permeabilization occurs.

Interestingly, the sequence of the dipeptides seemed to have an influence on the bacterial activity. Indeed, even though the reverse peptide **14** had a similar activity against Gram-positive bacteria as the one of peptide **12,** its potency against some of the Gram-negative strains was significantly lower. This decreased activity was accompanied by a higher hydrophobicity according to its longer retention time on reversed-phase HPLC (Figure 5). We anticipate that since the chemical composition of these peptides is similar, these different behaviors are likely related to a different spatial arrangement of the cationic and hydrophobic side-chains, giving a different amphiphilicity to peptide **14** vs. **12**. Indeed, AMPs usually adopt facially amphiphilic conformations in which cationic hydrophilic and hydrophobic side chains segregate onto opposite regions of the molecular surface. The importance of this overall topology and not the precise sequence, secondary structure, or chirality of the peptides has been highlighted as key features for their cell-killing activity [34]. Seminal works from Seebach [11] and more recently from Balaram [13,35] suggest that achiral β^2,2^-amino acids are β-turn inducers. In order to get some insight into the solution structure of these peptides, ^1^H NMR studies were conducted in D_2_O. Assignment of the proton signals was achieved by combination of COSY, TOCSY, and NOESY measurements. The data reveal that for peptides **9**, **11,** and **12**, one of the two β-protons C*H_2_*NH of the β^2,2^-*h*bis-Arg is significantly down-field shifted (2.4, 1.8 ppm, and 2.8 ppm respectively for **9**, **11,** and **12**) compared to the other (3.5 ppm), which is not the case for peptide **14**. Moreover, the presence of the *t*Bu group on the indole moiety has an important effect on the chemical shift of this proton since for peptides **8** and **10**, the chemical shift of this proton is 3.1–3.2 ppm. Altogether, these data suggest a close proximity between the β-protons C*H_2_*NH of the β^2,2^-*h*bis-Arg and the indole moiety in peptides **9**, **11**, and **12**, most likely because of cation-π interactions. Regardless of its nature, this specific conformation might favor the interaction of the peptide with the bacterial membrane and bring an explanation for the different biological behaviors of the two isomers **12** and **14** towards Gram-negative bacteria.

Regarding hemolysis (Figure 6), significant hemolytic effect was observed only at concentrations much higher than the antibacterial MIC values for the four most active peptides **11**–**14**, indicating a good selectivity of the compounds for bacterial cells over mammalian cells. Moreover, no cytotoxicity was observed for the 4 peptides **11**–**14** on human SHSYS5 cells. Finally, while introduction of fluorenyl or naphtyl group led to improved antibacterial activity for peptides **17**–**19**, this enhancement was, however, accompanied by a decreased selectivity on bacteria, and a significant increase in hemolysis and cytotoxicity on human cells (Table 1). We then evaluated the influence of an additional hydrophobic group on the C-terminal end (peptides **20**–**22**). While replacement of methyl ester with benzyl ester (**20**) or benzamide (**21**) gave peptides with enhanced efficiency, the incorporation of an alkyl chain (**22**) completely abolished the antimicrobial activity, probably reflecting an inappropriate balance between hydrophobicity and charge in this peptide.

Altogether, we selected peptide **11** as the best candidate for further analysis of its potential as therapeutic agent, thanks to its lack of haemolytic and cytolytic activity on mammalian cells and the easier synthesis of β^2,2^-*h*-bis-Orn-OH compared to β^3,3^-*h*-bis-Orn-OH. Since the incorporation of β-amino acids into peptides is known to improve their metabolic stability, the serum stability of this compound was first evaluated in human plasma (See Figure 10), where it appeared to be completely stable over 24 h, as expected for β-amino acids containing peptides sequences compared to a positive control peptide (4NGG) that was fully degraded in 20 mn (See 4. Materials and Methods).

The potency of peptide **11** was finally assessed in vivo in septic mice. In order to analyze its potential, mice were subjected to the acute model of sepsis “high grad sepsis” in which less than 50% of the mice survived to the procedure (See SI). In our technical conditions, 100% of the CLP-induced control mice succumbed during the five days following the induction of sepsis (Figure 9). However, the mice treated with one peritoneal injection of the peptide at 1 µg/g show a significant increase of the survival rate. Indeed, 50% of the mice treated with peptide **11** survived the acute peritonitis. The results revealed that the injection of peptide **11** induced an increase in the survival rate of CLP-treated mice.

Finally, this study validates these polycationic residues as new tools for the design of short bioactive antimicrobial cationic peptides. These new unnatural arginine analogs might be useful tools for other applications for which cationic residues are a key player, such as cell-penetrating peptides or RNA ligands.

## 4. Materials and Methods 

### 4.1. General Considerations

All reactions were carried out under argon atmosphere with dry commercial or freshly distilled solvents under anhydrous conditions unless otherwise stated. All reagents were purchased from commercial suppliers and used without further purification. Flash chromatography was performed using silica gel Merck 60 (0.040–0.063 μm, Molsheim, France). Analytical thin-layer chromatography (TLC) was performed using silica gel Merck 60 on alumina, visualized by UV fluorescence at 254 nm, and revealed with ninhydrin (0.3% in *n*-butanol/AcOH) or phosphomolybdic acid (solution in EtOH).

### 4.2. Solid Phase Peptide Synthesis

All reactions were carried out in Polypropylene Torviq syringes (sizes 5, 10, 20, or 50 mL) equipped with a porous polypropylene disc at the bottom and closed with an appropriate cap. HMBA resin (4-(Hydroxymethyl)benzoyl-aminoethyl) polystyrene (200–400 mesh, 0.8–1.2 mmol/g) was purchased from Iris Biotech (Marktredwitz, Germany). The loading of the Fmoc amino acid coupled resin was determined using a Cary3 U*v*/*v*IS spectrometer (Agilent, Santa-Clara, CA, USA). *O*-(Benzotriazol-1-yl)-*N,N,N′,N′*-tetramethyluronium hexafluorophosphate (HBTU) and 2-(1H-9-azabenzotriazole-1-yl)-1,1,3,3-tetramethyluronium hexafluorophosphate (HATU) were purchased from Iris Biotech. Solvents were purchased from VWR in HPLC grade and used without further purification. Purifications were performed by reverse-phase HPLC either on a Waters preparative HPLC system connected to a Breeze software (Fisher Scientific, Illkirch, France), using a *Waters XBridge* column (RP C18, 19 × 50 mm, 5 μm, 135 Å) at a flow rate of 14 mL/min or a Dionex semi-preparative HPLC-system connected to a Chromeleon software (Fisher Scientific, Illkirch, France), using a C18 semi-preparative column from AIT at a flow rate of 5 mL/min; and using as eluent A, H_2_O containing 0.1% of TFA, and as eluent B, CH_3_CN containing 0.1% of TFA. UV detection was done at 220 nm and 280 nm. Purification gradients were chosen to get a ramp of approximately 1% solution B per minute in the interest area. Peptide fractions from purification were analyzed by analytical HPLC, pooled according to their purity, partly concentrated under vacuum, and freeze-dried on an Alpha 2/4 freeze dryer from Bioblock Scientific (Fisher Bioblock Scientific, Rungis, France) to get the expected peptide as a powder. 

### 4.3. Product Characterisation

NMR spectra were recorded on Bruker ARX 250 (Bruker, France SAS, Wissembourg, France) or Brucker Avance III 300 spectrometers (Bruker, France SAS, Wissembourg, France) unless otherwise noted. Proton chemical shifts values (δ) are reported in parts per million (ppm) downfield from tetramethylsilane (TMS) unless noted otherwise. Coupling constants (*J*) are reported in Hertz (Hz). Carbon chemical shifts values (δ) are reported in parts permillion (ppm) with reference to internal solvent CDCl_3_ (77.00 ppm) or CD_3_OD (49.00 ppm). Multiplicities are abbreviated as follows: Singlet (s), doublet (d), triplet (t), quartet (q), multiplet (m), and broad singlet (bs). Signal assignments were made using COSY and HSQC experiments, and for peptides NOESY (250 ms mixing time), TOCSY (80 ms mixing time), and DQF-COSY spectra. High-resolution mass spectra (HRMS) were obtained on a Finnigan MAT 95 instrument and are given as experimental (found) and theoretical (calcd). Analytical RP-HPLC were performed on either a Waters system connected to a Breeze software or a Dionex system connected to a Chromeleon software. Waters system consisted of a binary pump (Waters 1525) and a dual wavelength U*v*/*v*isible Absorbance detector (Waters 2487, Saint-Quentin-en-Yveline, France). Dionex system consisted in an analytical automated LC system (Ultimate 3000) equipped with an auto sampler, a pump block composed of two ternary gradient pumps, and a dual wavelength detector. The analyses were performed on C18 analytical columns (from AIT (Paris, France) or Higgins (San Diego, CA, USA)) using as eluent A, H_2_O containing 0.1% of TFA and as eluent B, CH_3_CN containing 0.1% of TFA, at a flow rate of 1 mL/min. UV detection was done at 220 and 280 nm. Peptides were characterized by MALDI-TOF MS (DE-Pro, PerSeptive Biosystems, Framingham, MA, USA) in positive ion reflector mode using the matrix α-cyano-4-hydroxy-cinnamic acid (CHCA). Peptide molecular weights were determined for the free amine and not for the TFA salts.

#### 4.3.1. Synthesis of H-β^2,2^
*h*bis-Orn(Boc)_2_OMe 1 and Fmoc β^2,2^
*h*bis-Orn(Boc)_2_OH 2 (Scheme 5)

*Methyl 2,4-dicyano-2-(2-cyanoethyl)butanoate***23**: Methyl 2-cyanoacetate (10 g, 100 mmol) was mixed with acrylonitrile (11.7 g, 220 mmol) in a three-necked round bottom flask equipped with a condenser and an addition funnel. Triethylamine (6.8 mL, 50 mmol) was added dropwise at 0 °C through the addition funnel. The reaction was stirred continuously and allowed to react overnight at rt. After confirming completion of the reaction by TLC, AcOEt was added. The organic layer was washed with 5% citric acid solution and brine, dried over MgSO_4_, filtered, and evaporated. The product precipitated overnight. The solid was washed with AcOEt and obtained as a pale yellow powder (19.27 g, 93% yield); **R_f_** (Cy/AcOEt, 1:1) = 0.47; ^1^H NMR (250 MHz, CDCl_3_) δ 3.87 (s, 3H, CO_2_C*H*_3_), 2.37–2.62 (m, 4H, C*H*_2_β), 2.31 (ddd, *J* = 15.5 Hz, 8.6 Hz, 6.8 Hz, 2H, C*H*_2_γ), 2.14 (ddd, *J* = 14.2 Hz, 8.6 Hz, 6.1 Hz, 2H, C*H*_2_γ); ^13^C NMR (75 MHz, CDCl_3_): δ 166.6 (C, C=O), 117.2 (2C, *C*≡Nγ), 116.1 (C, *C*≡N α), 54.30 (CH_3_, CO_2_*C*H_3_), 47.6 (C, *C*α), 32.1 (2CH_2_, *C*H_2_β), 13.6 (2CH_2_, *C*H_2_γ); MS-ESI+: calcd for C_10_H_11_N_3_O_2_ 205.09, calcd for C_10_H_11_N_3_O_2_Na 228.08, found 228.07 [M + Na]^+^.

*Methyl 2-cyano-4-(Boc)amine-2-(3-(Boc)amine propyl)pentanoate***4**: Compound **23** (10 g, 49 mmol) was dissolved in methanol (25 mL). Boc_2_O (23.5 g, 108 mmol) and PtO_2_ (2.2 g, 9.8 mmol) were added and the reaction mixture was stirred at rt for 3 days under 5 bars of H_2_ pressure. The reaction mixture was filtered through a celite pad and evaporated to dryness. The crude compound was purified by flash chromatography (Cy/AcOEt 100:0 → 70:30) to afford yellowish oil (5 g, 21% yield); **R_f_** (Cy/AcOEt, 1:1) = 0.56; ^1^H NMR (300 MHz, CDCl_3_): δ (ppm) 4.69 (bs, 2H, N*H*), 3.76 (s, 3H, CO_2_C*H*_3_), 3.08–3.20 (m, 4H, C*H*_2_δ), 1.44–2 (m, 8H, C*H*_2_βανδC*H*_2_γ), 1.37–1.50 (m, 18H, C(C*H*_3_)_3_).; ^13^C NMR (62.5 MHz, CDCl_3_) δ 169.1 (C, C=O ester), 155.8 (2C, C=O carbamate), 118.7 (C, *C*≡N), 79.3 (2C, *C*(CH_3_)_3_), 53.3 (CH_3_, CO_2_*C*H_3_), 49.1 (C, *C*α), 39.7 (2CH_2_, *C*H_2_δ), 34.4 (2CH_2_, *C*H_2_β), 28.2 (6CH_3_, C(*C*H_3_)_3_), 26.1 (2CH_2_, *C*H_2_γ).; MS-ESI+: calcd for C_20_H_35_N_3_O_6_ 413.25, calcd for C_20_H_35_N_3_O_6_Na 436.24, found 436.24 [M + Na]^+^.

*H-β^2,2^ h*bis*-Orn(Boc)_2_OMe***1**: Compound **4** (2.35 g, 4.9 mmol) was dissolved in methanol (100 mL). Raney nickel was added, and the mixture was stirred under 5 bars of H_2_ pressure at rt for 3 days. The reaction mixture was filtered through a celite pad and evaporated to dryness. The product was used in peptide synthesis without further purification. (2.0 g, 98% yield); **R_f_** (Cy/AcOEt, 1:1) = 0.56; ^1^H NMR (300 MHz, MeOD) δ 3.68 (s, 3H, CO_2_C*H*_3_), 3.01 (t, *J* = 6.7 Hz, 4H, C*H*_2_δ), 2.77 (s, 2H, C*H*_2_βε), 1.55–1.60 (m, 4H, C*H*_2_β), 1.32–1.43 (m, 22H, C*H*_2_γ and C(C*H*_3_)_3_); ^13^C NMR (75 MHz, MeOD) δ 177.9 (C, C=O ester), 158.6 (2C, C=O carbamate), 79.9 (2C, *C*(CH_3_)_3_), 52.4 (CH_3_, CO_2_*C*H_3_), 51.5 (C, *C*α), 45.9 (CH_2_, *C*H_2_βε), 41.6 (2CH_2_, *C*H_2_δ), 31.2 (2CH_2_, *C*H_2_β), 28.8 (6CH_3_, C(*C*H_3_)_3_), 25.4 (2CH_2_, *C*H_2_γ); HRMS-ESI+: calcd for C_20_H_39_N_3_O_6_ 417.2839, found 418.2915 [M + H]^+^.

*Fmoc β^2,2^ h*bis*-Orn(Boc)_2_OH***2**: Compound **4** (2.3 g, 5.6 mmol) was dissolved in methanol (125 mL). An aqueous solution of sodium hydroxide (2 M) (12.5 mL, 25 mmol) and Raney nickel were added. The mixture was stirred under 5 bars of H_2_ pressure at rt for 7 days. The reaction mixture was filtered through a celite pad and evaporated to dryness. The crude compound was dissolved in a 1:1 mixture of THF and water (150 mL). FmocOSu (2.3 g, 6.8 mmol) and K_2_CO_3_ (1.7 g, 12.2 mmol) were added. The solution was allowed to react at rt overnight. After confirming the completion of the reaction by TLC, THF was evaporated. The resulting aqueous solution was acidified to pH = 2 by dropwise addition of 1M hydrochloric acid at 0 °C. The product was extracted with AcOEt, dried over MgSO_4_, filtered, and concentrated *in vacuo*. The crude compound was purified by flash chromatography (Cy/AcOEt/AcOH 100:0:1 → 75:25:1) to afford a white powder (2.5 g, 72% yield); **R_f_** (Cy/AcOEt/AcOH, 7:3:0.1) = 0.27; ^1^H NMR (250 MHz, CDCl_3_) δ 7.75 (d, *J* = 7.2 Hz, 2H, C*H* Ar), 7.58 (d, *J* = 7.2 Hz, 2H, C*H* Ar), 7.19–7.39 (m, 4H, C*H* Ar), 5.54 (bs, 1H, N*H* Fmoc), 4.95 (bs, 2H, N*H Boc*), 4.40 (d, *J* = 6.5 Hz, 2H, C*H*_2_ Fmoc), 4.20 (t, *J* = 6.5 Hz, 1H, C*H* Fmoc), 3.38–3.41 (m, 2H, C*H*_2_βε), 3.07 (m, 4H, C*H*_2_δ), 1.20–1.67 (m, 26H, C*H*_2_β, C*H*_2_γ and C(C*H*_3_)_3_); ^13^C NMR (62.5 MHz, CDCl_3_) δ 176.5 (C, C=O acid), 157.20, 156.4 (3C, C=O carbamate), 143.9, 141.3 (4C, *C* Ar), 129.1, 128.2, 127.7, 127.1, 125.3, 125.1, 120.0 (8CH, *C*H Ar), 79.3 (2C, *C*(CH_3_)_3_), 67.0 (CH_2_, *C*H_2_ Fmoc), 49.8 (CH_2_, *C*H_2_βε), 47.2 (CH, *C*H Fmoc), 40.7 (2CH_2_, *C*H_2_δ), 40.6 (C, *C*α), 30.6 (2CH_2_, *C*H_2_β), 28.4 (6CH_3_, C(*C*H_3_)_3_), 24.3 (2CH_2_, *C*H_2_γ); HRMS-ESI+: calcd for C_34_H_47_N_3_O_8_ 625.3255, calcd for C_34_H_47_N_3_O_8_Na 648,3153, found 648.3261 [M + Na]^+^.

#### 4.3.2. Synthesis of Boc-β^3,3^
*h*bis-Orn(Boc)_2_OH **3** (Scheme 6)

*1-Benzyl 3-*tert*-butyl 2,2-bis(2-cyanoethyl)malonate***24**: Benzyl *tert*-butylmalonate (25 g, 96.4 mmol) was mixed with acrylonitrile (14 mL, 210 mmol). Triethylamine (5.3 mL, 40 mmol) was added dropwise, followed by lithium perchlorate (5.4 g, 50 mmol). The reaction was stirred continuously and allowed to react overnight. After confirming completion of the reaction by TLC, AcOEt was added to the reaction mixture. The organic layer was washed with 5% citric acid solution and brine, dried over MgSO_4_, filtered, and evaporated. The crude compound was purified by flash chromatography (Cy/AcOEt, 100:0 to 8:2) to afford a yellow oil (30.2 g, 88% yield); **R_f_** (Cy/AcOEt, 8:2) = 0.37; ^1^H NMR (250 MHz, CDCl_3_) δ 7.37 (m, 5H, C*H* Ar), 5.20 (s, 2H, C*H*_2_Ph), 2.17–2.43 (m, 8H, C*H*_2_βανδC*H*_2_γ), 1.35 (s, 9H, C(C*H*_3_)_3_); ^13^C NMR (62.5 MHz, CDCl_3_) δ 169.2, 167.9 (2C, C=O), 134.5 (C, *C* Ar), 128.9, 128.8, 128.5, 127, 126.2 (5CH, *C*H Ar), 118.5 (2C, *C*≡N), 84 (C, *C*(CH_3_)_3_), 67.9 (CH_2_, *C*H_2_Ph), 56.2 (C, *C*α), 29.5 (2CH_2_, *C*H_2_β), 27.6 (3CH_3_, C(*C*H_3_)_3_), 13 (2CH_2_, *C*H_2_γ); HRMS-ESI+: calcd for C_20_H_24_N_2_O_4_ 356.1736, calcd for C_20_H_24_N_2_O_4_Na 379,1634, found 379.1628 [M + Na]^+^.

*2-(Tert-butoxycarbonyl)-4-cyano-2-(2-cyanoethyl)butanoic acid***5**: Compound **24** (18 g, 51 mmol) was dissolved in MeOH (500 mL). Ammonium formate (16.7 g, 265 mmol) and Pd/C (5.1 g, 100 mg/mmol) were added and the reaction mixture was stirred for 3 h. Afterward, the reaction mixture was filtered through a celite pad to remove the Pd/C before evaporation to dryness. The product was diluted with dichloromethane. The organic layer was washed with 10% citric acid solution and brine, dried over MgSO_4_, filtered, and evaporated to afford an oil. The product was used in the following step without further purification. (11.34 g, 83% yield); **R_f_** (Cy/AcOEt/AcOH, 8:2:0.1) = 0.1; ^1^H NMR (250 MHz, CDCl_3_) δ 2.41–2.54 (m, 4H, C*H*_2_β), 2.20 (t, *J* = 7.5 Hz, 4H, C*H*_2_γ), 1.51 (s, 9H, C(C*H*_3_)_3_); ^13^C NMR (62.5 MHz, CDCl_3_) δ 173.1 (C, C=O acid), 168.3 (C, C=O ester), 118.5 (2C, *C*≡N), 84.8 (C, *C*(CH_3_)_3_), 56.2 (C, *C*α), 30.0 (2CH_2_, *C*H_2_β), 27.8 (3CH_3_, C(*C*H_3_)_3_), 13.1 (2CH_2_, *C*H_2_γ); HRMS-ESI+: calcd for C_13_H_18_N_2_O_4_ 266.1267, calcd for C_13_H_18_N_2_O_4_Na 289,1165, found 289.1159 [M + Na]^+^.

*1-Tert-butyl 4-methyl 2,2-bis(2-cyanoethyl)succinate***6**: Compound **5** (9.0 g, 34 mmol) was dissolved in DCM under Argon. 1-Chloro-*N*,*N*-2-trimethylpropenylamine (9.0 mL, 68 mmol) was added. The solution was stirred for 2 h then concentrated *in vacuo*. The residue was dissolved in dry acetonitrile (170 mL) and cooled to 0 °C. DIEA (11.9 mL, 68 mmol) and a 2M solution of trimethylsilyldiazomethane in Et_2_O (34 mL, 68 mmol) was added. The reaction mixture was stirred at 0 °C for 16 h. The organic solvents were evaporated *in vacuo*. The residue was dissolved in AcOEt and washed with 10% citric acid, saturated NaHCO_3_ and brine. Finally, the organic layer was dried over MgSO_4_, filtered, and evaporated to dryness. The crude compound was dissolved in DMF (180 mL) and MeOH (90 mL) then Ag_2_O (39.4 g, 170 mmol) was added. The reaction mixture was refluxed for 10 min. After evaporation of MeOH, diethyl ether and a saturated solution of NH_4_Cl were added slowly and the mixture was filtered through a celite pad. The organic layer was separated and washed with a saturated solution of NH_4_Cl, dried over MgSO_4_, filtered, and evaporated. The crude compound was purified by flash chromatography (Cy/AcOEt, 100:0 to 60:40) to afford a yellow oil (2.1 g, 21% yield); **R_f_** (Cy/AcOEt, 1:1) = 0.6; ^1^H NMR (300 MHz, CDCl_3_) δ 3.71 (s, 3H, CO_2_C*H*_3_), 2.61 (s, 2H, C*H*_2_α), 2.27–2.37 (m, 4H, C*H*_2_γ), 1.91–2.07 (m, 4H, C*H*_2_δ), 1.48 (s, 9H, C(C*H*_3_)_3_); ^13^C NMR (75 MHz, CDCl_3_) δ 171.4, 170.1 (2C, C=O), 118.8 (2C, *C*≡N), 82.5 (C, *C*(CH_3_)_3_), 51.6 (CH_3_, CO_2_*C*H_3_), 46.6 (CH_2_, *C*H_2_α), 37.4 (C, *C*β), 30.6 (2CH_2_, *C*H_2_γ), 27.5 (3CH_3_, C(*C*H_3_)_3_), 12.3 (2CH_2_, *C*H_2_δ); HRMS-ESI+: calcd for C_15_H_22_N_2_O_4_ 294.1580, calcd for C_15_H_22_N_2_O_4_Na 317,1478, found 317.4718 [M + Na]^+^.

*2,2-Bis(2-cyanoethyl)-4-methoxy-4-oxobutanoic acid***25**: Compound **6** (1.3 g, 4.4 mmol) was dissolved in DCM (40 mL). Triisopropylsilane (900 µL, 4.4 mmol) and TFA (40 mL) were added. The reaction mixture was stirred for 1 hour before evaporation to dryness. The crude compound was purified by flash chromatography (Cy/AcOEt/AcOH, 100:0:1 to 50:50:1) to afford a colorless oil (900 mg, 86% yield); **R_f_** (Cy/AcOEt/AcOH, 5:5:0.1) = 0.34; ^1^H NMR (300 MHz, CDCl_3_) δ 3.66 (s, 3H, CO_2_C*H*_3_), 2.65 (s, 2H, C*H*_2_α), 2.30–2.49 (m, 4H, C*H*_2_γ), 1.98–2.16 (m, 4H, C*H*_2_δ); ^13^C NMR (75 MHz, CDCl_3_) δ 177.9 (C, C=O acid), 170.5 (C, C=O ester), 118.7 (2C, *C*≡N), 52.4 (CH_3_, CO_2_*C*H_3_), 46.3 (C, *C* β), 37.0 (CH_2_, *C*H_2_α), 30.8 (2CH_2_, *C*H_2_γ), 12.8 (2CH_2_, *C*H_2_δ); HRMS-ESI+: calcd for C_11_H_14_N_2_O_4_ 238.0954, calcd for C_11_H_14_N_2_O_4_Na 261,0852, found 261.0845 [M + Na]^+^.

*Methyl 3-((tert-butoxycarbonyl)amino)-5-cyano-3-(2-cyanoethyl)pentanoate***7**: Compound **25** (450 mg, 1.9 mmol) was dissolved in dry acetone (15 mL) and cooled to 0° C. NEt_3_ (300 µL, 2.3 mmol) and ClCO_2_Et (200 µL, 2.1 mmol) were added. The reaction mixture was stirred for 1.5 h. A solution of NaN_3_ (309 mg, 4.75 mmol) in H_2_O (8.5 mL) was added and the mixture was stirred at 0 °C for 2 additional hours. Acetone was evaporated and the compound was extracted with toluene. The organic layer was dried over MgSO_4_ and filtered. The volume was reduced by evaporation to 20 mL. *tert*-BuOH (15 mL) was added and the reaction was refluxed for 16 h. The solvent was evaporated and the crude compound purified by flash chromatography (Cy/AcOEt, 100:0 to 7:3) to afford a white powder (200 mg, 35% yield); **R_f_** (Cy/AcOEt, 1:1) = 0.44; ^1^H NMR (300 MHz, CDCl_3_) δ 5.17 (bs, 1H, N*H*Boc), 3.71 (s, 3H, CO_2_C*H*_3_), 2.60 (s, 2H, C*H*_2_α), 2.22–2.45 (m, 6H, C*H*_2_γ and C*H*_2_δ_1_), 1.99–2.12 (m, 2H, C*H*_2_δ_2_), 1.40 (s, 9H, C(C*H*_3_)_3_); ^13^C NMR (75 MHz, CDCl_3_) δ 170.4 (C, C=O ester), 154.3 (C, C=O carbamate), 119.2 (2C, *C*≡N), 80.5 and 80.4 (2C, *C*(CH_3_)_3_), 55.3 (C, *C*β), 52.4 (CH_3_, CO_2_*C*H_3_), 39.5 (CH_2_, *C*H_2_α), 31.8 (2CH_2_, *C*H_2_γ), 28.3 (3CH_3_, C(*C*H_3_)_3_), 12.0 (2CH_2_, *C*H_2_δ); HRMS-ESI+: calcd for C_15_H_23_N_3_O_4_ 309.1689, calcd for C_15_H_23_N_3_O_4_Na 332,1587, found 332.1581 [M + Na]^+^.

*Methyl-3,6-bis((tert-butoxycarbonyl)amino)-3-(3-((tert-butoxycarbonyl)amino)propyl) hexanoate***26**: Compound **7** (145 mg, 0.47 mmol) was dissolved in a 9:1 mixture of methanol and chloroform. PtO_2_ (16 mg, 0.07 mmol) was added and the reaction mixture was stirred under 5 bars of H_2_ pressure at rt for 3 days. The reaction mixture was filtered through a celite pad and evaporated to dryness. The crude product was dissolved in a 1:1 mixture of THF/H_2_O and Boc_2_O was added. After stirring overnight, THF was evaporated and the product was extracted with DCM. The organic layer was washed with brine, dried over MgSO_4_, filtered, and evaporated *in vacuo*. The crude compound was purified by flash chromatography (Cy/AcOEt, 7:3) to afford a colorless oil (150 mg, 62% yield); **R_f_** (Cy/AcOEt, 1:1) = 0.68; ^1^H NMR (300 MHz, CDCl_3_) δ 4.85 (bs, 1H, N*H* Boc), 4.72 (bs, 2H, N*H* Boc), 3.64 (s, 3H, CO_2_C*H*_3_), 3.02–3.11 (m, 4H, C*H*_2_ε), 2.61 (s, 2H, C*H*_2_α), 1.56–1.77 (m, 4H, C*H*_2_γ), 1.34–1.48 (m, 31H, C*H*_2_δ and C(C*H*_3_)_3_); ^13^C NMR (75 MHz, CDCl_3_) δ 171.7 (C, C=O ester), 156.0 (2C, C=O carbamate), 154.5 (C, C=O carbamate), 79.1 (3C, *C*(CH_3_)_3_), 56.1 (C, *C*β), 51.6 (CH_3_, CO_2_*C*H_3_), 40.5 (2CH_2_, *C*H_2_ε), 40.3 (CH_2_, *C*H_2_α), 33.3 (2CH_2_, *C*H_2_β), 28.4 (9CH_3_, C(*C*H_3_)_3_), 23.8 (2CH_2_, *C*H_2_γ); HRMS-ESI+: calcd for C_25_H_47_N_3_O_8_ 517.3363, calcd for C_25_H_47_N_3_O_8_Na 540,3261, found 540.3255 [M + Na]^+^.

*Boc-β^3,3^ h*bis*-Orn(Boc)_2_OH***3***: Compound***26** (0.130 g, 0.25 mmol) was dissolved in a 1:1 mixture of THF/H_2_O. LiOH (12 mg, 0.5 mmol) was added and the reaction mixture was stirred at rt for 5 days. THF was evaporated and the resulting aqueous solution was acidified to pH = 2 by dropwise addition of 1 M hydrochloric acid at 0 °C. The product was extracted with DCM and the organic layer was washed with brine, dried over MgSO_4_, filtered, and evaporated to afford a white powder (120 mg, 98% yield). The product was used in following step without any further purification; **R_f_** (Cy/AcOEt, 1:1) = 0.20; ^1^H NMR (300 MHz, MeOD) δ 3.01 (t, *J* = 6.6 Hz, 4H, C*H*_2_ε), 2.62 (s, 2H, C*H*_2_α), 1.73 (m, 4H, C*H*_2_γ), 1.42 (m, 31H, C*H*_2_δ and C(C*H*_3_)_3_); ^13^C NMR (75 MHz, MeOD) δ 158.6 (3C, C=O carbamate), 80.0 (3C, *C*(CH_3_)_3_), 57.2 (C, *C*β), 41.7 (3CH_2_, C*H*_2_α and C*H*_2_ε), 34.3 (2CH_2_, C*H*_2_γ), 29.0 (9CH_3_, C(*C*H_3_)_3_), 25.0 (2CH_2_, C*H*_2_δ); HRMS-ESI+: calcd for C_24_H_45_N_3_O_8_ 503.3207, calcd for C_24_H_45_N_3_O_8_Na 526,3105, found 526.3099 [M + Na]^+^; IR (ATR) υ (cm^−1^): 3346.9 (-OH acid), 2962.3, 2975.8, 2872.9, 2495.0, 1686.5 (C=O acid), 1514.6, 1479.5, 1453.5, 1392.0, 1365.3, 1273.7, 1248.7, 1162.3, 1092.6, 985.2, 866.5, 778.8.

#### 4.3.3. Synthesis of Fmoc-Tbt-OH (Scheme 7)

*(S)-2-((((9H-Fluoren-9-yl)methoxy)carbonyl)amino)-3-(2,4,6-tri-tert-butyl-1H-indol-3-yl) propanoic Acid*: A mixture of H-Trp-OH (3 g, 14.6 mmol) and *tert*-BuOH (31 mL, 323 mmol) in TFA (90 mL) was stirred at rt for 20 days. The resulting dark solution was evaporated to dryness to give a black oil, and water (50 mL) was added. To the resulting suspension was added KHCO_3_ until pH = 8–9. THF (50 mL) and FmocOSu (5.4 g, 16.0 mmol) were added and the mixture was stirred for 16 h. THF was evaporated and the solution was acidified to pH = 2. The compound was extracted with AcOEt, dried over MgSO_4_, filtered, and concentrated *in vacuo*. The crude compound was purified by flash chromatography (Cy/AcOEt/AcOH 100:0:1 to 50:50:1) to afford a white powder (6 g, 70% yield); **R_f_** (Cy/AcOEt/AcOH, 5:5:0.1) = 0.66; ^1^H NMR (300 MHz, CDCl_3_) δ 7.12–8.08 (m, 10H, C*H* Ar), 4.65–4.88 (m, 1H, C*H* Fmoc), 4.22–4.43 (m, 2H, C*H*_2_ Fmoc), 4.17 (t, *J* = 6.8 Hz, 1H, C*H*α), 3.56-3.74 (m, 1H, C*H*_2_β_1_), 3.42 (dd, *J* = 14.8 Hz, 9.1 Hz, 1H, C*H*_2_β_2_), 1.57 (s, 18H, C(C*H*_3_)_3_), 1.45 (s, 9H, C(C*H*_3_)_3_); ^13^C NMR (75 MHz, CDCl_3_) δ 177.7 (C, C=O acid), 156.1 (C, C=O carbamate), 143.8, 143.7, 142.9, 142.7, 141.2, 132.0, 130.2, 129.8 (9C, *C* Ar), 127.6, 127.0, 125.2, 125.1, 119.8, 116.9, 111.6 (10C, *C*H Ar), 103.9 (C, *C* Ar), 67.2 (CH_2_, *C*H_2_ Fmoc), 55.3 (CH, *C*Hα), 47.0 (CH, *C*H Fmoc), 34.8 (2C, *C*(CH_3_)_3_), 33.1 (C, *C*(CH_3_)_3_), 32.0, 30.9, 30.6 (9CH_3_, C(*C*H_3_)_3_), 27.6 (CH_2_, C*H*_2_β); HRMS-ESI+: calcd for C_38_H_46_N_2_O_4_ 594.3458, calcd for C_38_H_46_N_2_O_4_Na 617.3356, found 617.3350 [M + Na]^+^.

#### 4.3.4. Synthesis of Peptide A: Arg-Tbt-Arg-NH_2_



Fmoc Rink Amide resin loaded at 0.43 mmol/g (162 mg, 0.07 mmol) was washed with DMF and allowed to swell in DMF for 15 min. Fmoc deprotection was achieved through treatment of the resin with a solution of 20% piperidine (*v*:*v*) in DMF (5 min, 3 times), followed by washing with NMP. Fmoc-Arg(Pbf)-OH (4 eq, 0.28 mmol, 182 mg) was dissolved in dry NMP and HATU (3.6 eq, 0.25 mmol, 95 mg) and DIEA (10 eq, 0.7 mmol, 130 µL) were added. The resulting solution was added to the resin and the mixture was stirred for 2 h then filtrated and washed with NMP. Removal of the Fmoc protecting group was achieved by treatment of the resin with 20% (*v*:*v*) piperidine in DMF (3 times for 5 min). The resin was washed with NMP. Fmoc-Tbt-OH (4 eq, 0.28 mmol, 166 mg) was dissolved in NMP (1.5 mL). HATU (3.6 eq, 0.25 mmol, 95 mg) and DIEA (10 eq, 0.7 mmol, 130 µL) were added. The solution was added to the resin and the coupling reaction was allowed to proceed for 1.5 h at room temperature. The solution was removed by filtration and the resin was washed with DMF. After removal of the Fmoc protective group (20% piperidine in DMF, 5 min, 3 times) and washing of the resin with NMP, a solution of Fmoc-Arg(Pbf)-OH (4 eq, 0.28 mmol, 182 mg), HATU (3.6 eq, 0.25 mmol, 95 mg), and DIEA (10 eq, 0.7 mmol, 130 µL) in NMP (2 mL) was added and the reaction mixture was stirred for 2 h then filtrated and washed with NMP. Simultaneous final deprotection and cleavage from the resin was achieved by treating the resin with a TFA/TIS/H_2_O cocktail (95:2.5:2.5, 3 mL) for 4 h. The crude peptide was precipitated through addition of cold diethyl ether. Purification by preparative RP-HPLC using a gradient of 15% to 90% MeCN in 30 min gives after lyophilisation peptide A as a white powder with a purity of >95%. MALDI-TOF: calcd for C_36_H_62_N_10_O_3_ 683, found 684.4 [M + H]^+^, 706.4 [M + Na]^+^, 722.4 [M + K]^+^; HPLC (Water/ACN (0.1% TFA); 15% to 100% ACN in 30 min): tr = 10.19 min.

#### 4.3.5. Synthesis of Peptide B, Tbt-Arg-OMe B (Scheme 8)

Boc-Tbt-OH (50 mg, 0.11 mmol) was dissolved in DMF. HBTU (42 mg, 0.11 mmol) and DIEA (40 µL, 0.22 mmol) were added and the mixture was stirred for 5 min before addition of H-Aργ(Pbf)OMe (52 mg, 0.11 mmol). The reaction mixture was stirred at room temperature for 5 h, then diluted with Et_2_O and washed with an aqueous saturated solution of NH_4_Cl. The organic layer was dried over MgSO_4_, filtered, and evaporated to dryness. The crude compound was purified by flash chromatography (Cy/AcOEt, 100:0 to 50:50) to afford the pure protected dipeptide as a white powder (80 mg, 80% yield). Treatment of this compound with a cocktail of TFA/TIS/H_2_O (95:2.5:2.5) for 4 h, followed by evaporation to dryness lead to peptide **B**, which was purified by preparative RP-HPLC using a gradient of 30% to 50% MeCN in 30 min. After lyophilisation, peptide **B** was obtained as white powder with purity >98%; ^1^H NMR (300 MHz, MeOD) δ 7.24 (s, 1H, C*H* indole), 7.11 (s, 1H, C*H* indole), 4.25 (t, *J* = 6, 1H, C*H*α Arg), 4.08 (t, *J* = 8.1, 1H, C*H*α Tbt), 3.42 (d, *J* = 8.1, 1H, C*H*_2_β Tbt), 3.39 (s, 3H, COOC*H*_3_), 3.09–3.15 (m, 2H, C*H*_2_δ Arg), 1.70–1.74 (m, 1H, C*H*_2_γ_1_ Arg), 1.44–1.57 (m, 3H, C*H*_2_γ_2_ and C*H*_2_β Arg), 1.54 (s, 9H, C(C*H*_3_)_3_), 1.50 (s, 9H, C(C*H*_3_)_3_), 1.37 (s, 9H, C(C*H*_3_)_3_); MALDI-TOF: calcd for C_30_H_50_N_6_O_3_ 542.4, found 543.2 [M + H]^+^, 565.2 [M + Na]^+^; HPLC (Water/ACN (0.1% TFA); 5% to 100% ACN in 30 min: tr = 15.15 min.

#### 4.3.6. Synthesis of Peptides **8**–**12** by LPPS (Scheme 9)

##### Synthesis of Trp-β^2,2^
*h*bis-Orn-OMe **8** and Gdm-Trp-β^2,2^
*h*bis-Arg-OMe **10**

*Boc Trp-β^2,2^ h*bis*-Orn(Boc)_2_-OMe***15:** Boc-Tbt-OH (60 mg, 0.2 mmol) was dissolved in DMF (6 mL). HBTU (76 mg, 0.2 mmol) and DIEA (80 µL, 0.4 mmol) were added and the mixture was stirred for 5 min before addition of H-β^2,2^
*h*bis-Orn(Boc)_2_OMe **1** (84 mg, 0.2 mmol). The reaction mixture was stirred at room temperature overnight, then diluted with Et_2_O and washed with an aqueous saturated solution of NH_4_Cl. The organic layer was dried over MgSO_4_, filtered, and evaporated to dryness. The crude compound was purified by flash chromatography (Cy/AcOEt, 70:30) to afford **15** as a white powder (140 mg, 99% yield). ^1^H NMR (300 MHz, MeOD) δ 7.71 (d, *J* = 7.8, 1H, C*H* Ar), 7.38 (d, *J* = 7.2, 1H, C*H* Ar), 7.19 (td, *J* = 7.2, 1.1, 1H, C*H* Ar), 7.13 (td, *J* = 7.8, 1.1, 1H, C*H* Ar), 7.06 (d, *J* = 2.1, 1H, C*H* Ar), 5.91 (br, 1H, N*H* Boc), 5.36 (br, 1H, N*H* Boc), 4.78 (br, 1H, C*H*α Trp), 4.72 (br, 1H, C*H*_2_β_1_ Trp), 4.52 (br, 1H, C*H*_2_β_2_ Trp), 3.66 (s, 3H, CO_2_C*H*_3_), 3.30–3.35 (m, 2H, C*H*_2_βε_1_β^2,2^*h*bis-Orn), 3.09–3.23 (m, 2H, C*H*_2_βε_2_β^2,2^*h*bis-Orn), 2.95–3.04 (m, 4H, C*H*_2_δβ^2,2^*h*bis-Orn), 1.52 (s, 27H, C(C*H*_3_) _3_), 1.42–1.15 (m, 8H, C*H*_2_βανδC*H*_2_γβ^2,2^*h*bis-Orn). 

*Trp-β^2,2^ h bis-Orn-OMe***8:** Compound **15** (70 mg, 0.1 mmol) was dissolved in DCM (∼0.4 M) and an equivalent volume of TFA/TIS/H_2_O (95:2.5:2.5). The mixture was stirred at rt for 1 h then evaporated to dryness. The crude product was purified by preparative RP-HPLC using a gradient of 10% to 50% MeCN in 30 min. After lyophilisation compound **8** was obtained as white powder with purity >95% (30 mg, 70% yield); ^1^H NMR (500 MHz, D_2_O) δ 7.68 (d, *J* = 8, 1H, C*H* Ar), 7.55 (d, *J* = 12.8, 1H, C*H* Ar), 7.33 (s, 1H, C*H* Ar), 7.30 (t, *J* = 8, 1H, C*H* Ar), 7.22 (t, *J* = 7.5, 1H, C*H* Ar), 4.42 (dd, *J* = 9.5, 6, 1H, C*H*α Trp), 3.66 (s, 3H, CO_2_C*H*_3_), 3.51 (d, *J* = 14.5, 1H, C*H*_2_βε_1_ β^2,2^
*h* bis-Arg), 3.41 (dd, *J* = 14.2, 6, 1H, C*H*_2_β_1_ Trp), 3.35 (dd, *J* = 14.2, 9.5, 1H, C*H*_2_β_2_ Trp), 3.15 (d, *J* = 14.5, 1H, C*H*_2_βε_2_ β^2,2^
*h*bis-Arg), 2.81 (t, *J* = 7.8, 1H, C*H*_2_δ_1_ β^2,2^
*h* bis-Arg), 2.71 (t, *J* = 7.8, 1H, C*H*_2_δ_2_ β^2,2^
*h* bis-Arg), 1.49–1.53 (m, 1H, C*H*_2_γ_1_ β^2,2^
*h* bis-Arg), 1.35–1.39 (m, 3H, C*H*_2_γ_1’_ and C*H*_2_γ_2_ β^2,2^
*h* bis-Arg), 1.24 (td, *J* = 13.2, 3.7, 1H, C*H*_2_β_1_ β^2,2^
*h* bis-Arg), 1.05–1.12 (m, 2H, C*H*_2_β_1′_ and C*H*_2_β_2_β^2,2^
*h*bis-Arg), 0.88–0.95 (m, 1H, C*H*_2_β_2′_β^2,2^
*h*bis-Arg); MALDI-TOF: calcd for C_21_H_33_N_5_O_3_ 403.3, calcd for C_21_H_33_N_5_O_3_Na 426.3, found 404.5 [M + H]^+^, 426.5 [M + Na]^+^, 442.5 [M + K]^+^; HPLC (Water/ACN (0.1% TFA); 5% to 100% ACN in 30 min): tr = 7.18 min (Figure 11).

*Gdm-Trp-β^2,2^ h*bis*-Arg-OMe***10:** Compound **15** (70 mg, 0.1 mmol) was dissolved in DCM (∼0.4 M) and an equivalent volume of TFA/TIS/H_2_O (95:2.5:2.5). The mixture was stirred at rt for 1.5 h then evaporated to dryness. The crude compound was dissolved in 6 mL of THF 1,3-Di-Boc-2-(trifluoromethylsulfonyl)guanidine (137 mg, 0.35 mmol) and NEt_3_ (60 µL, 0.4 mmol) were added and the reaction mixture was stirred at rt overnight. After evaporation of THF, a solution of TFA/TIS/H_2_O (95:2.5:2.5) was added and the mixture was stirred at rt for 2 h. The crude product was purified by preparative RP-HPLC using a gradient of 10% to 50% MeCN in 30 min. After lyophilisation, compound **10** was obtained as white powder with purity >98% (31 mg, 57% yield); ^1^H NMR (300 MHz, D_2_O) δ 7.69 (d, *J* = 7.5, 1H, C*H* Ar), 7.36 (d, *J* = 8.1, 1H, C*H* Ar), 7.31 (s, 1H, C*H* Ar), 7.26 (td, *J* = 7.5, 0.9, 1H, C*H* Ar), 7.22 (td, *J* = 7.2, 0.9, 1H, C*H* Ar), 4.62 (t, *J* = 7.5, 1H, C*H*α Trp), 3.69 (s, 3H, CO_2_C*H*_3_), 3.47 (d, *J* = 14.1, 1H, C*H*_2_βε_1_ β^2,2^
*h*bis-Arg), 3.35 (d, *J* = 7.5, 2H, C*H*_2_β Trp), 3.21 (d, *J* = 14.4, 1H, C*H*_2_βε_2_ β^2,2^
*h*bis-Arg), 3.05 (t, *J* = 6.6, 1H, C*H*_2_δ_1_ β^2,2^
*h* bis-Arg), 2.98 (dd, *J* = 11.7, 6.6, 1H, C*H*_2_δ_2_ β^2,2^
*h* bis-Arg), 1.29–1.43 (m, 4H, C*H*_2_γ β^2,2^
*h*bis-Arg), 1.09-1.28 (m, 4H, C*H*_2_β β^2,2^
*h*bis-Arg); MALDI-TOF: calcd for C_32_H_57_N_5_O_3_ 529.3, calcd for C_32_H_57_N_5_O_3_Na 552.3, found 530.6 [M + H]^+^, 552.6 [M + Na]^+^, 513.6 [M + H − NH_3_]^+^; HPLC (Water/ACN (0.1% TFA); 5% to 100% ACN in 30 min): tr = 9.62 min (Figure 12).

##### Synthesis of Tbt-β^2,2^
*h*bis-Orn-OMe **9**, Tbt-β^2,2^ h bis-Arg-OMe **11** and Gua-Tbt-β^2,2^ h bis-Arg-OMe **12**


*Fmoc-Tbt*-*β^2,2^ h*bis*-Orn(Boc)_2_OMe***16:** Fmoc-Tbt-OH (400 mg, 0.64 mmol) was dissolved in DMF (24 mL). HBTU (244 mg, 0.64 mmol) and DIEA (240 µL, 1.28 mmol) were added and the mixture was stirred for 3 h before addition of H-β^2,2^
*h* bis-Orn(Boc)_2_OMe **1** (268 mg, 0.64 mmol). The reaction mixture was stirred at room temperature overnight, then diluted with Et_2_O and washed with an aqueous saturated solution of NH_4_Cl. The organic layer was dried over MgSO_4_, filtered, and evaporated to dryness. The crude compound was purified by flash chromatography (Cy/AcOEt, 100:0 to 70:30) to afford the pure protected dipeptide as a white powder (450 mg, 70% yield). ^1^H NMR (300 MHz, MeOD) δ 8.22 (s, 1H, N*H* indole), 7.75 (d, *J* = 7.2, 1H, C*H* Ar Fmoc), 7.55 (d, *J* = 7.2, 1H, C*H* Ar Fmoc), 7.41 (s, 1H, C*H* Ar indole), 7.35 (t, *J* = 7.2, 1H, C*H* Ar Fmoc), 7.24 (dt, *J* = 11.7 and 7.2, 1H, C*H* Ar Fmoc), 7.12 (s, 1H, C*H* indole), 4.27–4.33 (m, 3H, C*H*α Tbt and C*H_2_* Fmoc), 4.12 (t, *J* = 6.9, 1H, C*H* Fmoc), 3.55 (s, 3H, CO_2_C*H*_3_), 3.43 (dd, *J* = 14.1 9.3, 1H, C*H*_2_β_1_ Tbt), 3.39 (d, *J* = 14.1, 1H, C*H*_2_βε_1_ β^2,2^
*h* bis-Orn), 3.23 (dd, *J* = 14.4, 6.3, 1H, C*H*_2_β_2_ Tbt), 2.9 (m, 4H, C*H*_2_δ β^2,2^
*h* bis-Orn), 2.79 (d, *J* = 14.1, 1H, C*H*_2_βε_2_ β^2,2^
*h*bis-Orn), 1.52 (s, 9H, C(C*H*_3_)_3_ indole), 1.47 (s, 9H, C(C*H*_3_)_3_ indole), 1.36–1.44 (m, 35H, C(C*H*_3_)_3_ indole, C(C*H*_3_)_3_ Boc, C*H*_2_β β^2,2^
*h*bis-Orn and C*H*_2_γ β^2,2^
*h*bis-Orn); ^13^C NMR (75 MHz, MeOD) δ 177.1 (C, C=O amide), 174.9 (C, C=O ester), 158.4 (C=O Boc), 157.9 (C=O Fmoc), 145.2, 145.1, 143.6, 142.9, 142.5, 133.1, 131.8, 131.3 (8C, *C* Ar), 128.7, 128.2, 126.2, 120.9 (4CH, *C*H Ar Fmoc), 117.3, 113.4 (2CH, *C*H Ar indole), 106.2 (C, *C* Ar), 79.8 (C, *C*(CH_3_)_3_ Boc), 68.1 (CH_2_, *C*H_2_ Fmoc), 58.5 (CH, *C*Hα Tbt), 52.3 (CH_3_, CO_2_*C*H_3_), 50.7 (C, *C*α β^2,2^
*h*bis-Orn), 48.3 (CH, *C*H Fmoc), 42.9 (CH_2_, *C*H_2_βε β^2,2^
*h*bis-Orn), 41.6 (CH_2_, *C*H_2_δ β^2,2^
*h*bis-Orn), 35.7, 35.5 and 34.3 (3C, *C*(CH_3_)_3_ indole), 32.7 (CH_3_, C(*C*H_3_)_3_ indole), 32.4 and 32.1 (CH_2_, *C*H_2_γ β^2,2^
*h* bis-Orn), 31.3 (CH_3_, C(*C*H_3_)_3_ indole), 30.9 (CH_3_, C(*C*H_3_)_3_ indole), 29 (CH_2_, *C*H_2_β Tbt), 28.8 (6CH_3_, C(*C*H_3_)_3_ Boc), 25.5 and 25.3 (CH_2_, *C*H_2_β β^2,2^
*h* bis-Orn).

*H-Tbt*-*β^2,2^ h bis-OrnOMe***9:** Compound **16** (60 mg, 0.06 mmol) was dissolved in a 20% solution of piperidine in DCM and allowed to react for 1 h before evaporation to dryness. A solution of TFA/TIS/H_2_O (95:2.5:2.5) was added and the mixture was stirred at rt for 1 h. The crude product was purified by preparative RP-HPLC using a gradient of 30% to 50% MeCN in 30 min. After lyophilisation **9** was obtained as white powder with a purity of 98% (20 mg, 58% yield); ^1^H NMR (300 MHz, D_2_O) δ 8.54 (s, 1H, N*H* indole), 7.26 (s, 1H, C*H* Ar), 7.22 (s, 1H, C*H* Ar), 4.03 (dd, *J* = 9.3, 6, 1H, C*H*α Tbt), 3.50 (s, 3H, CO_2_C*H*_3_), 3.37 (d, *J* = 14.2, 1H, C*H*_2_βε_1_ β^2,2^
*h* bis-Orn), 3.34 (d, *J* = 14.1, 2H, C*H*_2_β Tbt), 2.80 (m, 4H, C*H*_2_δ β^2,2^
*h* bis-Orn), 2.33 (d, *J* = 14.2, 1H, C*H*_2_βε_2_ β^2,2^
*h* bis-Orn), 1.31–1.44 (m, 35H, C*H*_2_β β^2,2^
*h* bis-Orn, C*H*_2_γ β^2,2^
*h* bis-Orn and C(C*H*_3_)_3_); MALDI-TOF: calcd for C_32_H_57_N_5_O_3_ 571.5, found 572.6 [M + H]^+^; HPLC (Water/ACN (0.1% TFA); 30% to 50% ACN in 30 min): tr = 12.73 min (Figure 13).

*H-Tbt*-*β^2,2^ h bis-Arg-OMe 11 and Gua-Tbt*-*β^2,2^ h bis-Arg-OMe***12:** Compound **9** (10 mg, 0.013 mmol) was dissolved in 1 mL of THF. 1,3-Di-Boc-2-(trifluoromethylsulfonyl) guanidine (30 mg, 0.08 mmol) and DIEA (27 µL, 0.156 mmol) were added and the reaction mixture was stirred at rt for 2 h. After evaporation of THF, a solution of TFA/TIS/H_2_O (95:2.5:2.5) was added and the mixture was stirred at rt for 2 h. The crude product was evaporated *in vacuo* and purified by preparative RP-HPLC using a gradient of 30% to 50% MeCN in 30 min. Two pics were collected separately at 14 and 18 min corresponding, respectively, to compounds **11** and **12**. After lyophilisation, the two compounds **11** (5 mg, 59% yield) and **12** (2 mg, 22% yield) were obtained as white powders with purity >99%.

*H-Tbt*-*β^2,2^ h bis-Arg-OMe***11**: ^1^H NMR (500 MHz, D_2_O) δ 7.29 (s, 1H, C*H* indole), 7.27 (s, 1H, C*H* indole), 4.14 (dd, *J* = 11.2, 5.5, 1H, *C*Hα Tbt), 3.61 (dd, *J* = 14.5, 5, 1H, C*H*_2_β_1_ Tbt), 3.57 (s, 3H, CO_2_C*H*_3_), 3.44 (dd, *J* = 13.5, 12, 2H, C*H*_2_β_2_ Tbt), 3.21 (d, *J* = 14.5, 1H, C*H*_2_βε_1_ β^2,2^
*h* bis-Arg), 3.04 (m, *J* = 4H, C*H*_2_δ β^2,2^
*h* bis-Arg), 1.83 (d, *J* = 14.5, 1H, C*H*_2_βε_2_ β^2,2^
*h* bis-Arg), 1.55 (s, 9H, C(C*H*_3_)_3_), 1.49 (s, 9H, C(C*H*_3_)_3_), 1.39 (s, 9H, C(C*H*_3_)_3_), 1.1-1.34 (m, 8H, C*H*_2_β β^2,2^
*h* bis-Arg and C*H*_2_γ β^2,2^
*h* bis-Arg); MALDI-TOF: calcd for C_35_H_61_N_5_O_3_ 655.5, calcd for C_35_H_61_N_5_O_3_Na 678.5, found 656.4 [M + H]^+^, 678.4 [M + Na]^+^, 694.4 [M + K]^+^, 639.4 [M + H − NH_3_]^+^; HPLC (Water/ACN (0.1% TFA); 30% to 50% ACN in 30 min: tr = 16.11 min (Figure 14).

*Gua-Tbt*-*β^2,2^ h bis-Arg-OMe***12**: ^1^H NMR (300 MHz, MeOD) δ 8.35 (s, 1H, N*H* indole), 7.31 (d, *J* = 1.5, 1H, C*H* Ar), 7.14 (d, *J* = 1.5, 1H, C*H* Ar), 4.39 (t, *J* = 7.2, 1H, C*H*α Tbt), 3.65 (s, 3H, CO_2_C*H*_3_), 3.52 (d, *J* = 14.2, 1H, C*H*_2_βε_1_ β^2,2^
*h* bis-Arg), 3.47 (dd, *J* = 11.1, 7.2, 2H, C*H*_2_β Tbt), 3.01–3.13 (m, 4H, C*H*_2_δ β^2,2^
*h* bis-Arg), 2.82 (d, *J* = 14.2, 1H, C*H*_2_βε_2_ β^2,2^
*h* bis-Arg), 1.26–1.64 (m, 35H, C*H*_2_β β^2,2^
*h* bis-Arg, C*H*_2_γ β^2,2^
*h* bis-Arg and C(C*H*_3_)_3_); MALDI-TOF: calcd for C_36_H_63_N_11_O_3_ 697.5, calcd for C_36_H_63_N_11_O_3_Na 720.5, found 698.4 [M + H]^+^, 720.4 [M + Na]^+^, 736.3 [M + K]^+^, 681.3 [M + H − NH_3_]^+^; HPLC (Water/ACN (0.1% TFA); 30% to 70% ACN in 30 min): tr = 14.08 min (Figure 15).

#### 4.3.7. Synthesis of Peptides 13 and 14 by SPPS (Scheme 10)

*H-β^3,3^-h-bis-Arg-Tbt-OMe***13**: HMBA-AM resin (108 mg, 0.1 mmol) was washed five times with DMF, DCM, and DMF, then allowed to swell in DMF for 30 min. Fmoc-Tbt-OH (4 eq, 0.4 mmol, 238 mg) was dissolved in dry DCM. The solution was cooled to 0 °C, DIC (4 eq, 0.4 mmol, 60 µL) was added. The reaction was stirred for 1.5 h and the solvent was then removed *in vacuo*. The resulting anhydride was dissolved in DMF and added to the resin. A solution of DMAP (0.1 eq, 0.04 mmol, 5 mg) in DMF was added and the resin was shaken for 1 h before washing with DMF, DCM, and DMF (resin loading = 0.84 mmol/g). Removal of the Fmoc protecting group was achieved by treatment of the resin with 20% (*v*:*v*) piperidine in DMF 3 times for 5 min. The resin was washed five times with DMF. Boc-β^3,3^*h* bis-Orn(Boc)_2_OH **3** (2 eq, 0.18 mmol, 90 mg) was dissolved in DMF (1.2 mL). HATU (1.8 eq, 0.17 mmol, 65 mg) and DIEA (2 eq, 0.18 mmol, 23 µL) were added. The solution was added to the resin (0.09 mmol, 108 mg) and the coupling reaction was allowed to proceed for 2 h at room temperature. The solution was removed by filtration and the resin was washed with DMF five times. Reaction completion was monitored by Kaiser test. Boc removal was performed by treating the resin with a TFA/TIS/H_2_O cocktail (95:2.5:2.5) for 5 h. The free amines were then reacted with 1,3-Di-Boc-2-(trifluoromethylsulfonyl)guanidine (10 eq, 1.8 mmol, 700 mg) and NEt_3_ (10 eq, 1.8 mmol, 240 µL) in DMF overnight. The resin was filtrated and the Boc groups were removed with a TFA/TIS/H_2_O cocktail (95:2.5:2.5) at rt for 3 h. After filtration, peptide **14** was cleaved from the resin using a mixture of MeOH/DMF/DIEA (5:5:1) for 16 h at 50 °C. The solution was filtrated, and solvents were evaporated. The crude product was purified by preparative RP-HPLC using a gradient of 30% to 70% MeCN in 30 min. After lyophilisation peptide **13** was obtained as a white powder with a purity of >95% (47 mg, 75% yield); ^1^H NMR (500 MHz, D_2_O) δ 7.37 (d, *J* = 1.7, 1H, C*H* Ar indole), 7.28 (d, *J* = 1.7, 1H, C*H* Ar indole), 4.69 (t, *J* = 7.5, 1H, C*H*α Tbt), 3.48–3.53 (m, 1H, C*H*_2_β_1_ Tbt), 3.68 (dd, *J* = 10.5, 1, 1H, C*H*_2_β_2_ Tbt), 3.25 (t, *J* = 6.5, 1H, C*H*_2_ε_1_ β^2,2^
*h* bis-Arg), 3.17 (t, *J* = 6.5, 1H C*H*_2_ε_2_ β^2,2^
*h* bis-Arg), 2.69 (d, *J* = 16, 1H, C*H*_2_α_1_ β^2,2^
*h* bis-Arg), 2.55 (d, *J* = 16, 1H, C*H*_2_α_2_ β^2,2^
*h* bis-Arg), 1.70–1.75 (m, 4H, C*H*_2_γ β^2,2^
*h* bis-Arg), 1.64-1.68 (m, 2H, C*H*_2_δ_1_ β^2,2^
*h* bis-Arg), 1.55–1.61 (m, 2H, C*H*_2_δ_2_ β^2,2^
*h* bis-Arg), 1.42 (s, 9H, C(C*H*_3_)_3_), 1.40 (s, 9H, C(C*H*_3_)_3_), 1.29 (s, 9H, C(C*H*_3_)_3_); MALDI-TOF: calcd for C_35_H_61_N_9_O_2_ 655.5, calcd for C_35_H_61_N_9_O_2_Na 678.5, found 656.5 [M + H]^+^, 678.5 [M + Na]^+^, 694.5 [M + K]^+^; HPLC (Water/ACN (0.1% TFA); 5% to 100% ACN in 30 min): tr = 19.29 min (Figure 16).

*Gdm-β^2,2^-h-bis-Arg-Tbt-OMe***14:** HMBA-AM resin (185 mg, 0.2 mmol) was washed five times with DMF, DCM. and DMF, then allowed to swell in DMF for 30 min. Fmoc-Tbt-OH (4 eq, 0.4 mmol, 238 mg) was dissolved in dry DCM. The solution was cooled to 0 °C, and DIC (4 eq, 0.4 mmol, 60 µL) was added. The reaction was stirred for 30 min and the solvent was then removed *in vacuo*. The resulting anhydride was dissolved in DMF and added to the resin. A solution of DMAP (0.1 eq, 0.04 mmol, 5 mg) in DMF was added and the resin was shaken for 1 h before washing with DMF, DCM, and DMF (resin loading = 0.4 mmol/g). Removal of the Fmoc protecting group was achieved by treatment of the resin with 20% (*v*:*v*) piperidine in DMF 3 times for 5 min. The resin was washed five times with DMF. Fmoc-β^2,2^
*h* bis-Orn(Boc)_2_OH **2** (3 eq, 0.22 mmol, 141 mg) was dissolved in DMF (1.5 mL). HATU (1.8 eq, 0.21 mmol, 80 mg) and DIEA (4 eq, 0.88 mmol, 150 µL) were added. The solution was added to the resin (0.07 mmol, 185 mg) and the reaction was allowed to proceed at 50 °C for 16 h. The solution was removed by filtration and the resin was washed with DMF five times. The reaction being incomplete as revealed by a Kaiser test, the same coupling procedure was repeated a second time. The resin was then treated with a 20% solution of piperidine in DMF for 5 min 3 times. Boc removal was performed by treatment with a TFA/TIS/H_2_O cocktail (95:2.5:2.5) for 1 h. The free amines were then reacted with 1,3-Di-Boc-2-(trifluoromethylsulfonyl)guanidine (5 eq, 0.35 mmol, 137 mg) and NEt_3_ (10 eq, 0.7 mmol, 90 µL) in DMF overnight. The resin was filtrated and the Boc groups were removed with a TFA/TIS/H_2_O cocktail (95:2.5:2.5) at rt for 3 h. After filtration, peptide **14** was cleaved from the resin using a mixture of MeOH/DMF/DIEA (5:5:1) for 16 h at 50 °C. The solution was filtrated, and solvents were evaporated. The crude product was purified by preparative RP-HPLC using a gradient of 40% to 90% MeCN in 30 min. After lyophilisation, peptide **14** was obtained as white powder with a purity of >95% (34 mg, 72% yield); ^1^H NMR (300 MHz, D_2_O) δ 7.36 (s, 1H, C*H* indole), 7.19 (s, 1H, C*H* indole), 4.7 (m, 1H, C*H*α Tbt), 3.61 (s, 3H, CO_2_C*H*_3_), 3.53 (dd, *J* = 15.3, 6.6, 1H, C*H*_2_β_1_ Tbt), 3.31–3.36 (m, 2H, C*H*_2_β_2_ Tbt and C*H*_2_βε_1_ β^2,2^
*h* bis-Arg), 2.88 (t, *J* = 7, 1H, C*H*_2_δ_1_ β^2,2^
*h* bis-Arg), 2.85 (t, *J* = 7, 1H, C*H*_2_δ_2_ β^2,2^
*h* bis-Arg), 2.83 (d, *J* = 14.5, 1H, C*H*_2_βε_2_ β^2,2^
*h* bis-Arg), 1.42 (s, 9H, C(C*H*_3_)_3_), 1.39 (s, 9H, C(C*H*_3_)_3_), 1.27 (s, 9H, C(C*H*_3_)_3_), 1.14–1.33 (m, 8H, C*H*_2_β and C*H*_2_γ β^2,2^
*h* bis-Arg); MALDI-TOF: calcd for C_36_H_63_N_11_O_3_ 697.5, calcd for C_36_H_63_N_11_O_3_Na 720.5, found 698.5 [M + H]^+^, 720.4 [M + Na]^+^, 736.4 [M + K]^+^; HPLC (Water/ACN (0.1% TFA); 30% to 70% ACN in 30 min): tr = 16.5 min (Figure 17).

#### 4.3.8. Synthesis of Fmoc-Tbt-β^2,2^-*h*-bis-Orn-OMe **17a** and Fmoc-Tbt-β^2,2^-*h*-bis-Arg-OMe **17b** (Scheme 11)

*Fmoc-Tbt*-*β^2,2^-h-bis-Orn-OMe***17a**: Compound **16** (42 mg, 0.042 mmol) was treated with a mixture of TFA/TIS/H_2_O (95:2.5:2.5, V = 1mL) at rt for 3 h and then evaporated to dryness. The crude product was purified by preparative RP-HPLC using a gradient of 50% to 100% MeCN in 30 min. After lyophilisation compound **17a** was obtained as white powder with purity >99% (27 mg, 95% yield); ^1^H NMR (300 MHz, CD_3_OD) δ 7.80 (d, *J* = 7.5 Hz, 2H, C*H* Fmoc), 7.60 (t, *J* = 8.8 Hz, 2H, C*H* Fmoc), 7.37–7.42 (m, 5H, C*H* indole and C*H* Fmoc), 7.13 (d, *J* = 1.4 Hz, 1H, C*H* indole), 4.44 (dt, *J* = 9.8 Hz, 7.7 Hz, 1H, C*H* Fmoc), 4.17–4.24 (m, 3H, C*H_2_* Fmoc and C*H*α Tbt), 3.59 (d, *J* = 14.2 Hz, 1H, C*H*_2_βε_1_ β^2,2^
*h* bis-Arg), 3.41 (dd, *J* = 14.6 Hz, 9 Hz, 1H, C*H*_2_β_1_ Tbt), 3.26 (dd, *J* = 14.2 Hz, 5.8 Hz, 1H, C*H*_2_β_2_ Tbt), 2.88 (d, *J* = 14.2 Hz, 1H, C*H*_2_βε_2_ β^2,2^
*h* bis-Arg), 2.79–2.82 (m, 4H, C*H*_2_δ β^2,2^
*h* bis-Arg), 1.22–1.69 (m, 35H, C(C*H*_3_)_3_, C*H*_2_β β^2,2^
*h* bis-Arg and C*H*_2_γ β^2,2^
*h* bis-Arg); **MALDI-TOF**: calcd for C_48_H_67_N_5_O_5_ 793.5, found 794.5 [M + H]^+^, 816.4 [M + Na]^+^, 832.4 [M + K]^+^; **HPLC** (Water/ACN (0.1% TFA); 50% to 100% ACN in 10 min: tr = 6.29 min (Figure 18).

##### Synthesis of Fmoc-Tbt-β^2,2^-*h*-bis-Arg-OMe **17b**

Compound **17a** (50 mg, 0.05 mmol) was dissolved 1 mL of THF. 1,3-Di-Boc-2-(trifluoromethylsulfonyl)guanidine (51 mg, 0.13 mmol) and NEt_3_ (35 µL, 0.26 mmol) were added and the reaction mixture was stirred at rt for 24 h. After evaporation of THF, a solution of TFA/TIS/H_2_O (95:2.5:2.5, 2 mL) was added and the mixture was stirred at rt for 45 min. The crude product was evaporated *in vacuo* and purified by preparative RP-HPLC using a gradient of 30% to 100% MeCN in 30 min. After lyophilisation compound **17b** was obtained as white powders with purity >99% (25 mg, 57% yield); ^1^H NMR (300 MHz, CD_3_OD) δ 7.80 (d, *J* = 7.5 Hz, 2H, C*H* arom Fmoc), 7.60 (d, *J* = 7.5 Hz, 2H, C*H* arom Fmoc), 7.39 (t, *J* = 7.5 Hz, 2H, C*H* arom Fmoc), 7.33 (s, *1*H, C*H* arom indole), 7.27 (t, *J* = 7.5 Hz, 2H, C*H* arom Fmoc), 7.13 (s, 1H, C*H* indole), 4.27–4.38 (m, 2H, C*H_2_* Fmoc), 4.18–4.22 (m, 2H, C*H* Fmoc and C*H*α Tbt), 3.56 (d, *J* = 11.8 Hz, 1H, C*H*_2_βε_1_ β^2,2^
*h* bis-Arg), 3.40 (dd, *J* = 14.6 Hz, 9.1 Hz, 1H, C*H*_2_β_1_ Tbt), 3.25 (dd, *J* = 14.6 Hz, 5.5 Hz, 1H, C*H*_2_β_2_ Tbt), 3–3.07 (m, *4*H, C*H*_2_δ β^2,2^
*h* bis-Arg), 2.72 (d, *J* = 14.3 Hz, 2H, C*H*_2_βε_2_ β^2,2^
*h* bis-Arg), 1.34–1.53 (m, 35H, C(C*H*_3_)_3_, C*H*_2_β β^2,2^
*h* bis-Arg and C*H*_2_γ β^2,2^
*h* bis-Arg); MALDI-TOF: calcd for C_50_H_71_N_9_O_5_ 877.6, found 878.4 [M + H]^+^, 916.4 [M + K]^+^; HPLC (Water/ACN (0.1% TFA); 5% to 100% ACN in 30 min: tr = 24.59 min (Figure 19).

#### 4.3.9. Synthesis of Fluo-Tbt-β^2,2^ h bis-Orn-OMe **18** (Scheme 12)

Compound **16** (43 mg, 0.043 mmol) was dissolved in THF (1 mL) and treated with DBU (0.2 µL, 0.0013 mmol) and octanethiol (75 µL, 0.43 mmol) at rt for 10 min and then evaporated to dryness. The crude compound was purified by flash chromatography (DCM/MeOH/NEt_3_, 100:0:0 to 80:20:1) affording a white powder (17 mg, 99% yield). The product was dissolved in 2 mL of THF. 2,7-di-*tert*-butylfluorène-9-carboxylic acid (17 mg, 0.05 mmol), HBTU (16 mg, 0.043 mmol), and DIEA (75 µL, 0.43 mmol) were added and the reaction mixture was stirred at rt overnight. After evaporation of THF, a solution of TFA/TIS/H_2_O (95:2.5:2.5, 2 mL) was added and the mixture was stirred at rt for 45 min and then evaporated. The crude product was purified by preparative RP-HPLC using a gradient of 50% to 100% MeCN in 30 min. After lyophilisation compound **18** was obtained as white powder with purity >99% (24 mg, 65% yield); ^1^H NMR (300 MHz, CD_3_OD) δ 7.79 (s, C*H* arom fluorenyl), 7.71 (d, *J* = 8 Hz, 2H, C*H* arom fluorenyl), 7.59 (s, C*H* arom fluorenyl), 7.49 (d, *J* = 8 Hz, 2H, C*H* arom fluorenyl), 7.36 (s, 1H, C*H* indole), 7.14 (s, 1H, C*H* indole), 4.38 (dd, *J* = 10 Hz, 4 Hz, 1H, C*H*α Tbt), 3.65 (d, *J* = 14 Hz, 1H, C*H*_2_βε_1_ β^2,2^
*h* bis-Arg), 3.56 (dd, *J* = 14.8 Hz, 10.5 Hz, 1H, C*H*_2_β_1_ Tbt), 3.39 (dd, *J* = 14.8 Hz, 4 Hz, 1H, C*H*_2_β_2_ Tbt), 2.65 (d, *J* = 14.3 Hz, 2H, C*H*_2_βε_2_ β^2,2^
*h* bis-Arg), 2.52–2.66 (m, *2*H, C*H*_2_δ_1_ β^2,2^
*h* bis-Arg), 2.38–2.45 (m, *2*H, C*H*_2_δ β^2,2^
*h* bis-Arg), 1.30–1.53 (m, 35H, C(C*H*_3_)_3_, C*H*_2_β β^2,2^
*h* bis-Arg and C*H*_2_γ β^2,2^
*h* bis-Arg); MALDI-TOF: calcd for C_55_H_81_N_5_O_4_ 875.6, found 876.6 [M + H]^+^, 898.6 [M + Na]^+^, 914.6 [M + K]^+^; HPLC (Water/ACN (0.1% TFA); 45% to 100% ACN in 30 min: tr = 21.9 min (Figure 20).

#### 4.3.10. Synthesis of Np-Tbt-β^2,2^ h bis-Orn-OMe **19** (Scheme 13)

Compound **16** (50 mg, 0.05 mmol) was dissolved in THF (1 mL) and treated with DBU (0.3 µL, 0.002 mmol) and octanethiol (90 µL, 0.5 mmol) at rt for 10 min and then evaporated to dryness. The crude compound was purified by flash chromatography (DCM/MeOH/NEt_3_, 100:0:0 to 80:20:1) affording a white powder (34 mg, 87% yield). This compound was dissolved in 4 mL of THF. 2-Naphtoyl chloride (9.5 mg, 0.05 mmol) and NEt_3_ (14 µL, 0.1 mmol) were added and the reaction mixture was stirred at rt overnight. After evaporation of THF, a solution of TFA/TIS/H_2_O (95:2.5:2.5, 2 mL) was added and the mixture was stirred at rt for 30 min and then evaporated. The crude product was purified by preparative RP-HPLC using a gradient of 30% to 100% MeCN in 30 min. After lyophilisation compound **19** was obtained as white powder with 96% purity (22 mg, 60% yield); ^1^H NMR (300 MHz, CD_3_OD) δ 8.06 (s, 1H, C*H* Np), 7.91 (d, *J* = 8.4 Hz, 2H, C*H* Np), 7.86 (d, *J* = 7.8 Hz, 1H, C*H* Np), 7.75 (d, *J* = 8.4 Hz, 1H, C*H* Np), 7.60 (t, *J* = 6.4 Hz, 1H, C*H* Np), 7.56 (t, *J* = 6.4 Hz, 1H, C*H* Np), 7.41 (s, 1H, C*H* indole), 7.18 (s, 1H, C*H* indole), 4.57 (t, *J* = 7.5 Hz, 1H, *C*Hα Tbt), 3.69 (d, *J* = 14.2, 2H, C*H*_2_βε_1_ β^2,2^
*h* bis-Arg), 3.63 (dd, *J* = 14.7 Hz, 8.3 Hz, 1H, C*H*_2_β_1_ Tbt), 3.48 (dd, *J* = 14.7 Hz, 6.6 Hz, 1H, C*H*_2_β_2_ Tbt), 2.99 (d, *J* = 14.2, 2H, C*H*_2_βε_1_ β^2,2^
*h* bis-Arg), 2.78–2.88 (m, 4H, C*H*_2_δ β^2,2^
*h* bis-Arg), 1.18–1.37 (m, 35H, C(C*H*_3_)_3_, C*H*_2_β β^2,2^
*h* bis-Arg and C*H*_2_γ β^2,2^
*h* bis-Arg); MALDI-TOF: calcd for C_44_H_65_N_5_O_5_ 725.5, found 726.4 [M + H]^+^, 748.4 [M + K]^+^, 764.4 [M + K]^+^; HPLC (Water/ACN (0.1% TFA); 5% to 100% ACN in 30 min: tr = 21.9 min (Figure 21).

#### 4.3.11. Synthesis of Tbt-β^2,2^ h bis-Arg-OBn **20** (Scheme 14)

*β^2,2^-h-bis-Orn(Boc)_2_OBn***27**: Fmoc β^2,2^-*h*-bis-Orn(Boc)_2_OH **2** (300 mg, 0.48 mmol) was dissolved in MeCN (1.7 mL). After addition of Cs_2_CO_3_ (188 mg, 0.58 mmol) and benzyl bromide (63 µL, 0.53 mmol), the reaction mixture was heated at 60 °C under microwave (150W) for 10 min. The solution was filtered and evaporated to dryness. The crude compound was dissolved in AcOEt and washed with an aqueous solution of NaHCO_3_ 5% followed by a solution of citric acid 5%, then dried over MgSO_4_, filtered, and concentrated *in vacuo*. The crude compound was purified by flash chromatography (DCM/MeOH/NEt_3_ 100:0:0.1 to 95:5:0.1) to afford a white powder (80 mg, 28% yield); ^1^H NMR (300 MHz, CD_3_OD) δ 7.34–7.43 (m, 5H, C*H* Ar), 5.17 (s, 2H, C*H*_2_Ph), 3.01 (t, *J* = 6.8 Hz, 4H, C*H*_2_δ), 2.84 (s, 2H, C*H*_2_βε), 1.62 (dd, *J* = 9.3 Hz, 5.3 Hz, 4H, C*H*_2_β), 1.43 (s, 18H, C(C*H*_3_)_3_), 1.24–1.41 (m, 4H, C*H*_2_γ); ^13^C NMR (75 MHz, CD_3_OD) δ 176.9 (C, C=O ester), 158.5 (2C, C=O carbamate), 137.5 (C, *C* Ar), 129.6, 129.4, 129.3 (3CH, *C*H Ar), 79.9 (2C, *C*(CH_3_)_3_), 67.5 (CH_2_, *C*H_2_Ph), 51.3 (CH_2_, *C*H_2_βε), 45.5 (C, *C*α), 41.5 (2CH_2_, *C*H_2_δ), 31.1 (2CH_2_, *C*H_2_β), 28.8 (6CH_3_, C(*C*H_3_)_3_), 25.3 (2CH_2_, *C*H_2_γ); HRMS-ESI+: calcd for C_26_H_43_N_3_O_6_ 493.3152, found 494.3225 [M + H]^+^.

*Fmoc-Tbt*-*β^2,2^-h-bis-Orn(Boc)_2_OBn***28:** Fmoc-Tbt-OH (83 mg, 0.14 mmol) was dissolved in DMF (6 mL). HBTU (53 mg, 0.14 mmol) and DIEA (24 µL, 0.14 mmol) were added and the mixture was stirred for 5 min before addition of H-β^2,2^-*h*-bis-Orn(Boc)_2_OBn **28** (70 mg, 0.14 mmol). The reaction mixture was stirred at room temperature overnight, then diluted with Et_2_O and washed with an aqueous saturated solution of NH_4_Cl. The organic layer was dried over MgSO_4_, filtered, and evaporated to dryness. The crude compound was purified by flash chromatography (Cy/AcOEt, 100:0 to 70:30) to afford the pure protected dipeptide as a white powder (67 mg, 45% yield). ^1^H NMR (300 MHz, MeOD) δ 7.91 (s, 1H, N*H* indole), 7.66 (d, *J* = 7.5, 1H, C*H* Ar Fmoc), 7.45 (d, *J* = 7.2, 1H, C*H* Ar Fmoc), 7.37 (s, 1H, C*H* Ar indole), 7.3 (t, *J* = 7.5, 1H, C*H* Ar Fmoc), 7.18 (dt, *J* = 14.7 and 6.9, 1H, C*H* Ar Fmoc), 7.09 (s, 1H, C*H* indole), 5.89 (bs, 2H, N*H* Boc), 5.71 (bs, 1H, N*H* Fmoc), 4.87 (s, 2H, C*H*_2_Ph), 4.20–4.34 (m, 3H, C*H*α Tbt and C*H_2_* Fmoc), 4.09 (t, *J* = 7.2, 1H, C*H* Fmoc), 3.37 (d, *J* = 14.1, 1H, C*H*_2_βε_1_ β^2,2^
*h* bis-Orn), 3.31 (d, *J* = 7.5 Hz, 2H, C*H*_2_β Tbt), 2.86 (m, 4H, C*H*_2_δ β^2,2^
*h* bis-Orn), 2.55 (d, *J* = 14.1, 1H, C*H*_2_βε_2_ β^2,2^
*h* bis-Orn), 1.52 (s, 9H, C(C*H*_3_)_3_ indole), 1.47 (s, 9H, C(C*H*_3_)_3_ indole), 1.36–1.44 (m, 35H, C(C*H*_3_)_3_ indole, C(C*H*_3_)_3_ Boc, C*H*_2_β β^2,2^
*h* bis-Orn and C*H*_2_γ β^2,2^
*h* bis-Orn).

*H-Tbt*-*β^2,2^ h bis-Arg-OBn***20:** Compound **28** (68 mg, 0.064 mmol) was dissolved in DCM (∼0.4 M) and an equivalent volume of TFA. The mixture was stirred at rt for 1.5 h then evaporated to dryness. The crude compound was dissolved in 4 mL of THF. 

1,3-Di-Boc-2-(trifluoromethylsulfonyl)guanidine (50 mg, 0.128 mmol) and NEt_3_ (500 µL, 3.2 mmol) were added and the reaction mixture was stirred at rt for 16 h. After evaporation of THF, the crude mixture was dissolved in a 20% solution of piperidine in DCM and allowed to react for 2 h before evaporation to dryness. A solution of TFA/TIS (95:5) was added and the mixture was stirred at rt for 1.5 h. After evaporation, the crude product was purified by preparative RP-HPLC using a gradient of 20% to 90% MeCN in 30 min. After lyophilisation compound **20** was obtained as white powder with purity >99% (10 mg, 22% yield); ^1^H NMR (300 MHz, CD_3_OD) δ 8.36 (s, 1H, N*H* indole), 7.20–7.32 (m, 5H, C*H* Ph), 7.28 (s, 1H, C*H* indole), 7.15 (s, 1H, C*H* indole), 5.03 (d, *J* = 12 Hz, 1H, C*H*_2_ Ph), 5.01 (d, *J* = 12 Hz, 1H, C*H*_2_ Ph), 4.03 (t, *J* = 6.1 Hz, 1H, *C*Hα Tbt), 3.60 (d, *J* = 14.2, 1H, C*H*_2_βε_1_ β^2,2^
*h* bis-Arg), 3.42 (d, *J* = 6.6 Hz, 2H, C*H*_2_β Tbt), 2.97–3 (m, 4H, C*H*_2_δ β^2,2^
*h* bis-Arg), 2.33 (d, *J* = 1.24 Hz, 1H, C*H*_2_βε_2_ β^2,2^
*h* bis-Arg), 1.20–1.59 (m, 35H, C(C*H*_3_)_3_, C*H*_2_β β^2,2^
*h* bis-Arg and C*H*_2_γ β^2,2^
*h* bis-Arg); MALDI-TOF: calcd for C_48_H_67_N_5_O_5_ 731.5, found 732.4 [M + H]^+^, 770.3 [M + K]^+^, 716.3 [M + H − NH_3_]^+^; HPLC (Water/ACN (0.1% TFA); 40% to 90% ACN in 10 min: tr = 7.51 min (Figure 22). 

#### 4.3.12. Synthesis of Tbt-β^2,2^ h bis-Arg-NHBn **21** (Scheme 15)

*β^2,2^-h-bis-Orn(Boc)_2_NHBn***29**: Fmoc β^2,2^-*h*-bis-Orn(Boc)_2_OH **2** (300 mg, 0.48 mmol) was dissolved in DCM (20 mL). DCC (100 mg, 0.48 mmol), HOBt (64 mg, 0.48 mmol), DMAP (5 mg, 0.05 mmol), and benzyl amine (56 mg, 0.53 mmol) were added. The reaction mixture was stirred at rt overnight. The solution was washed with an aqueous saturated solution of NaCl, dried over MgSO_4_, filtered, and concentrated *in vacuo*. The crude compound was purified by flash chromatography (Cy/AcOEt 100:0 to 50:50) to afford a colorless oil (251 mg, 73% yield); **R_f_** (Cy/AcOEt, 7:3) = 0.5; ^1^H NMR (300 MHz, CDCl_3_) δ 7.76 (d, *J* = 7.5 Hz, 2H, C*H* Ar), 7.58 (d, *J* = 7.5 Hz, 2H, C*H* Ar), 7.40 (t, *J* = 7.5 Hz, 2H, C*H* Ar), 7.25–7.33 (m, 7H, C*H* Ar), 6.48 (bs, 1H, N*H* amide), 5.47 (bs, 1H, N*H* Fmoc), 4.78 (bs, 2H, N*H* Boc), 4.36–4.42 (m, 4H, C*H*_2_Ph, C*H*_2_ Fmoc), 4.15–4.20 (m, 1H, C*H* Fmoc), 3.36–3.38 (m, 2H, C*H*_2_βε), 2.98–3.04 (m, 4H, C*H*_2_δ), 1.31–1.69 (m, 26H, C*H*_2_β, C(C*H*_3_)_3_, C*H*_2_γ); ^13^C NMR (75 MHz, CD_3_OD) δ 175.3 (C, C=O amide), 157.3 (C, C=O carbamate), 156.3 (C, C=O carbamate), 143.9, 141.4, 138.3 (3C, *C* Ar), 128.8, 127.8, 127.6, 127.1, 125.1, 120.1 (6CH, *C*H Ar), 79.2 (C, *C*(CH_3_)_3_), 66.9 (CH_2_, *C*H_2_ Fmoc), 49.8 (C, *C*α), 47.3 (CH, *C*H Fmoc), 44.6 (CH_2_, *C*H_2_βε), 43.8 (CH_2_, *C*H_2_Ph), 40.7 (CH_2_, *C*H_2_δ), 30.8 (CH_2_, *C*H_2_β), 29.0 (CH_3_, C(*C*H_3_)_3_), 24.2 (CH_2_, *C*H_2_γ); Fmoc β^2,2^-*h*-bis-Orn(Boc)_2_NHBn (251 mg, 0.35 mmol) was dissolved in THF (6 mL). Octanethiol (600 µL, 3.5 mmol) and DBU (1.5 µL, 0.01 mmol) were added. The reaction mixture was stirred for 15 min then concentrated *in vacuo*. The crude compound was purified by flash chromatography (DCM/MeOH/NEt_3_ 100:0:0 to 80:20:0.1) to afford **29** as a colorless oil (150 mg, 87% yield); ^1^H NMR (300 MHz, CDCl_3_) δ 8.94 (bs, *2*H, N*H_2_*), 7.15–7.24 (m, 5H, C*H* Ar), 4.75 (bs, 2H, N*H* Boc), 4.33–4.35 (m, 2H, C*H*_2_Ph), 2.98–3.11 (m, 4H, C*H*_2_δ), 2.80 (s, 2H, C*H*_2_βε), 1.01–1.73 (m, 26H, C*H*_2_β, C(C*H*_3_)_3_, C*H*_2_γ); ^13^C NMR (75 MHz, CD_3_OD) δ 176.3 (C, C=O amide), 156.1 (C, C=O carbamate), 139 (C, *C* Ar), 128.5, 127.3, 127 (3CH, *C*H Ar), 78.9 (C, *C*(CH_3_)_3_), 47.3 (C, *C*α), 45.3 (CH_2_, *C*H_2_βε), 42.9 (CH_2_, *C*H_2_Ph), 40.8 (CH_2_, *C*H_2_δ), 31.6 (CH_2_, *C*H_2_β), 28.4 (CH_3_, C(*C*H_3_)_3_), 22.4 (CH_2_, *C*H_2_γ); HRMS-ESI+: calcd for C_26_H_44_N_4_O_5_ 492.3312, found 515.3199 [M + Na]^+^.

*Fmoc-Tbt*-*β^2,2^ h bis-Orn(Boc)_2_NHBn***30**: Fmoc-Tbt-OH (90 mg, 0.15 mmol) was dissolved in DMF (6 mL). HBTU (57 mg, 0.15 mmol) and DIEA (30 µL, 0.15 mmol) were added and the mixture was stirred for 5 min before addition of H-β^2,2^*h* bis-Orn(Boc)_2_NHBn **29** (75 mg, 0.15 mmol). The reaction mixture was stirred at room temperature overnight, then diluted with Et_2_O and washed with an aqueous saturated solution of NH_4_Cl. The organic layer was dried over MgSO_4_, filtered, and evaporated to dryness. The crude compound was purified by flash chromatography (Cy/AcOEt, 100:0 to 60:40) to afford the pure protected dipeptide **30** as a white powder (64 mg, 40% yield). ^1^H NMR (300 MHz, MeOD) δ 7.91 (s, 1H, N*H* indole), 7.65 (d, *J* = 7.5, 2H, C*H* Ar Fmoc), 7.40 (d, *J* = 7.4, 2H, C*H* Ar Fmoc), 7.35 (s, 1H, C*H* Ar indole), 7.28 (t, *J* = 7.4, 2H, C*H* Ar Fmoc), 7.03–7.20 (m, 8H, C*H* Ar Fmoc, C*H* indole, C*H* benzyl), 6.35 and 6.19 (2 bs, 1H, N*H* Amide), 5.61 (bs, 1H, N*H* Fmoc), 4.69 and 4.76 (2bs, 2H, N*H* Boc), 4.15–4.27 (m, 5H, C*H*α Tbt, C*H_2_* Fmoc, C*H*_2_Ph), 4.08 (t, *J* = 76.8, 1H, C*H* Fmoc), 3.24–3.39 (m, 3H, C*H*_2_β_1_ Tbt, C*H*_2_βε β^2,2^
*h* bis-Orn), 2.84–2.94 (m, 4H, C*H*_2_δ β^2,2^
*h* bis-Orn), 2.67–2.74 (m, 1H, C*H*_2_β_2_ Tbt), 1.30–1.34 (m, 53H, C(C*H*_3_)_3_ indole, C(C*H*_3_)_3_ Boc, C*H*_2_β β^2,2^
*h* bis-Orn and C*H*_2_γ β^2,2^
*h* bis-Orn) ^13^C NMR (75 MHz, CD_3_OD) δ 175.1 and 172.4 (C, C=O amide), 156.2 (C, C=O carbamate), 143.9, 143.6, 142.8, 142.6, 141.3, 141.2, 138.3, 132.1, 130.2, 129.8 (10C, *C* Ar), 128.7, 127.7, 127.4, 127.1, 125.3, 125.2, 120, 116.8, 112.1 (9CH, *C*H Ar), 104.5 (C, *C* indolyl), 79.1 (C, *C*(CH_3_)_3_), 67.2 (CH_2_, *C*H_2_ Fmoc), 56.7 (CH, *C*Hα Tβτ), 49.2 (C, *C*α β^2,2^
*h* bis-Orn), 47.1 (CH, *C*H Fμοχ), 43.6 (CH_2_, *C*H_2_Ph), 42.9 (*C*H_2_, *C*H_2_β Tbt), 40.7 (2CH_2_, *C*H_2_δ β^2,2^
*h* bis-Orn), 34.9, 34.7, 33.1 (3CH_2_, *C*H_2_β ανδ *C*H_2_β’ β^2,2^
*h* bis-Orn), 32.2, 30.9, 30.8, 30.7 (4CH_3_, C(*C*H_3_)_3_), 24.2, 24.3 (2CH_2_, *C*H_2_γ).

*H-Tbt*-*β^2,2^ h bis-Arg-NHBn***21:** Compound **30** (64 mg, 0.06 mmol) was dissolved in DCM (∼0.4 M) and an equivalent volume of TFA containing 5% of TIS. The mixture was stirred at rt for 2 h then evaporated to dryness. The crude compound was dissolved in 4 mL of THF. 1,3-Di-Boc-2-(trifluoromethylsulfonyl)guanidine (117 mg, 0.3 mmol) and NEt_3_ (80 µL, 0.6 mmol) were added and the reaction mixture was stirred at rt for 48 h. After evaporation of THF, the crude mixture was dissolved in a 20% solution of piperidine in DCM and allowed to react for 2 h before evaporation to dryness. A solution of TFA/TIS (95:5) in DCM was added and the mixture was stirred at rt for 1.5 h. After evaporation, the crude product was purified by preparative RP-HPLC using a gradient of 40% to 100% MeCN in 30 min. After lyophilisation, compound **21** was obtained as white powder with purity >99% (20 mg, 45% yield); ^1^H NMR (300 MHz, CD_3_OD) δ 7.32 (s, 1H, C*H* indole), 7.25–7.32 (m, 5H, C*H* arom), 7.17 (s, 1H, C*H* indole), 4.28 (s, 2H, C*H*_2_Ph), 4.06 (dd, *J* = 9.4 Hz, 6.1 Hz, 1H, *C*Hα Tbt), 3.44–3.56 (m, 3H, C*H*_2_βε_1_ β^2,2^
*h* bis-Arg and C*H*_2_β Tbt), 3.11–3.17 (m, 4H, C*H*_2_δ β^2,2^
*h* bis-Arg), 2.39 (d, *J* = 14 Hz, 1H, C*H*_2_βε_2_ β^2,2^
*h* bis-Arg), 1.22–1.54 (m, 35H, C(C*H*_3_)_3_, C*H*_2_β β^2,2^
*h* bis-Arg and C*H*_2_γ β^2,2^
*h* bis-Arg); MALDI-TOF: calcd for C_41_H_66_N_10_O_2_ 730.5, found 731.6 [M + H]^+^; HPLC (Water/ACN (0.1% TFA); 40% to 100% ACN in 30 min: tr = 13.57 min (Figure 23).

#### 4.3.13. Synthesis of Tbt-β^2,2^ h bis-Arg-NH(CH_2_)_13_CH_3_
**22** (Scheme 16)

*β^2,2^-h-bis-Orn(Boc)_2_NH(CH_2_)_13_CH_3_***31**: Fmoc β^2,2^*h* bis-Orn(Boc)_2_OH **2** (300 mg, 0.48 mmol) was dissolved in DCM (25 mL). DCC (109 mg, 0.53 mmol), HOBt (72 mg, 0.53 mmol), DMAP (5 mg, 0.05 mmol), and tetradecyl amine (113 mg, 0.53 mmol) were added. The reaction mixture was stirred at for 4 h. The solution was washed with brine, dried over MgSO_4_, filtered, and concentrated *in vacuo*. The crude compound was purified by flash chromatography (Cy/AcOEt 100:0 to 50:50) to afford a colorless oil (390 mg, 99% yield); **R_f_** (Cy/AcOEt, 1:1) = 0.76; ^1^H NMR (300 MHz, CDCl_3_) δ 7.67 (d, *J* = 7.3 Hz, 2H, C*H* Ar), 7.51 (d, *J* = 7.3 Hz, 2H, C*H* Ar), 7.31 (t, *J* = 7.3 Hz, 2H, C*H* Ar), 7.22 (t, *J* = 7.3 Hz, 2H, C*H* Ar), 6.05 (bs, 1H, N*H* amide), 5.65 and 5.47 (2bs, 1H, N*H* Fmoc), 4.80 (bs, 2H, N*H* Boc), 4.30 (d, *J* = 6.2 Hz, 2H, C*H*_2_ Fmoc), 4.11 (t, *J* = 6.9 Hz, 1H, C*H* Fmoc), 3.28 (d, *J* = 6.1 Hz, 1H, C*H*_2_βε β^2,2^
*h* bis-Orn), 3.13 (m, 2H, C*H*_2_C_13_H_27_), 3 (m, 4H, C*H*_2_δ β^2,2^
*h* bis-Orn), 1.41 (m, 4H, C*H*_2_γ β^2,2^
*h* bis-Orn), 1.17-1.35 (m, 46H, C(C*H*_3_)_3_ Boc, (C*H_2_*)_12_CH_3_ and C*H*_2_β β^2,2^
*h* bis-Orn), 0.8 (t, 3H, (CH_2_)_12_C*H*_3_); ^13^C NMR (75 MHz, CD_3_OD) δ 175.2 (C, C=O amide), 157.3 and 156.3 (2C, C=O carbamate), 143.9 and 141.3 (C, *C* arom Fmoc), 127.7, 127.1, 125.1, 120 (5CH, *C*H arom Fmoc), 79.1 (C, *C*(CH_3_)_3_), 66.9 (CH_2_, *C*H_2_ Fmoc), 49.6 (C, *C*α β^2,2^
*h* bis-Orn), 47.2 (C, *C*H Fmoc), 44.6 (CH_2_, *C*H_2_βε β^2,2^
*h* bis-Orn), 40.7 (CH_2_, C*H*_2_C_13_H_27_), 39.8 (CH_2_, *C*H_2_δ β^2,2^
*h* bis-Orn), 33.9 (CH_2_, C*H*_2_C_12_H_25_), 31.9 (CH_2_, *C*H_2_β β^2,2^
*h* bis-Orn), 30.8 (CH_2_, C*H*_2_C_11_H_23_), 29.72, 29.68, 29.62, 29.59, 29.39, 29.33 (CH_2_, C*H*_2_C_7_H_17_), 28.4 (CH_3_, C(*C*H_3_)_3_), 27.1, 27.0, 24.2 (CH_2_, (*C*H_2_)_3_CH_3_), 22.7 (CH_2_, *C*H_2_γ β^2,2^
*h* bis-Orn), 14.2 (CH_3_, (CH_2_)_13_*C*H_3_); HRMS-ESI+: calcd for C_48_H_76_N_4_O_7_ 820.5714, found 843.5606 [M + Na]^+^. The obtained Fmoc β^2,2^-*h*-bis-Orn(Boc)_22_NH(CH_2_)_13_CH_3_ (130 mg, 0.12 mmol) was dissolved in THF (2 mL). Octanethiol (210 µL, 1.2 mmol) and DBU (0.5 µL, 0.0036 mmol) were added. The reaction mixture was stirred for 20 min then concentrated *in vacuo*. The crude compound was purified by flash chromatography (DCM/MeOH/NEt_3_ 100:0:0.1 to 80:20:0.1) to afford a colorless oil (58 mg, 81% yield); ^1^H NMR (300 MHz, CD_3_OD) δ 3.21 (t, *J* = 7.2 Hz, 2H, C*H*_2_C_12_H_27_), 33.04 (t, *J* = 6.9 Hz, 4H, C*H*_2_δ β^2,2^
*h* bis-Orn), δ 2.78 (s, 2H, C*H*_2_β’ β^2,2^
*h* bis-Orn), 1.32-1.58 (m, 48H, C(C*H*_3_)_3_ Boc, CH_2_(C*H_2_*)_11_CH_3_, C*H*_2_β β^2,2^
*h* bis-Orn and C*H*_2_γ β^2,2^
*h* bis-Orn), 0.93 (t, 3H, *J* = 6.6 Hz, (CH_2_)_13_C*H*_3_); ^13^C NMR (75 MHz, CD_3_OD) δ 178.2 (C, C=O amide), 159.4 (C, C=O carbamate), 79.8 (C, *C*(CH_3_)_3_), 50.2 (C, *C*α), 45.8 (CH_2_, *C*H_2_β’), 41.7 (CH_2_, *C*H_2_δ), 40.4 (CH_2_, *C*H_2_(CH_2_)_12_CH_3_), 33.1, 31.6, 30.8, 30.5 (4CH_2_), 28.8, (6CH_3_, C(*C*H_3_)_3_), 28.2, 25.3, 23.7 (CH_2_), 14.5 (CH_3_, CH_2_*C*H_3_); HRMS-ESI+: calcd for C_33_H_66_N_4_O_5_ 598.5033, found 599.5112 [M + H]^+^.

*Fmoc-Tbt*-*β^2,2^-h-bis-Orn(Boc)_2_NH(CH_2_)_13_CH_3_***32**: Fmoc-Tbt-OH (54 mg, 0.09 mmol) was dissolved in DMF (5 mL). HBTU (34 mg, 0.09 mmol) and DIEA (20 µL, 0.09 mmol) were added and the mixture was stirred for 5 min before addition of β^2,2^-*h*-bis-Orn(Boc)_2_NH(CH_2_)_13_CH_3_ 31 (55 mg, 0.09 mmol). The reaction mixture was stirred at room temperature for 4 h, then diluted with Et_2_O and washed with an aqueous saturated solution of NH_4_Cl. The organic layer was dried over MgSO_4_, filtered, and evaporated to dryness. The crude compound was purified by flash chromatography (Cy/AcOEt, 100:0 to 50:50) to afford the pure protected dipeptide as a white powder (70 mg, 66% yield). ^1^H NMR (300 MHz, CDCl_3_) δ 7.99 (s, 1H, N*H* indole), 7.23 (d, *J* = 7.5, 2H, C*H* Ar Fmoc), 7.50 (d, *J* = 7.2, 2H, C*H* Ar Fmoc), 7.43 (s, 1H, C*H* Ar indole), 7.38 (t, *J* = 7.5, 2H, C*H* Ar Fmoc), 7.24 (t, *J* = 7.5, 2H, C*H* Ar Fmoc), 7.17 (s, 1H, C*H* indole), 6.23 (bs, 1H, N*H* amide), 5.92 and 5.66 (2bs, 1H, N*H* Fmoc), 4.78 and 4.86 (2bs, 2H, N*H* Boc), 4.25–4.37 (m, 3H, C*H*α Tbt and C*H_2_* Fmoc), 4.13–4.17 (m, 1H, C*H* Fmoc), 3.33–3.44 (m, 3H, C*H*_2_βε β^2,2^
*h* bis-Orn and C*H*_2_β_1_ Tbt), 3.02 (m, 6H, C*H*_2_δ β^2,2^
*h* bis-Orn and C*H*_2_(CH_2_)_12_CH_3_), 2.75–2.80 (m, 1H, C*H*_2_β_2_ Tβτ), 1.30–1.47 (m, 77H, 3C(C*H*_3_)_3_ indole, CH_2_(C*H_2_*)_12_CH_3_, 2C(C*H*_3_)_3_ Boc, 2C*H*_2_β β^2,2^
*h* bis-Orn and 2C*H*_2_γβ^2,2^*h* bis-Orn), 0.86 (t, *J* = 6.9, 3H, (CH_2_)_13_C*H*_3_) ^13^C NMR (75 MHz, CDCl_3_) δ 174.8 and 172.3 (C, C=O amide), 156.1 and 156.2 (C, C=O carbamate), 143.9, 143.6, 142.8, 142.6, 141.3, 141.2, 132.1, 130.2, 129.8 (9C, *C* Ar), 127.7, 127.1, 125.3, 125.2, 120, 116.8, 112.1 (7CH, *C*H Ar), 104.5 (C, *C* indolyl), 79.1 (C, *C*(CH_3_)_3_), 67.2 (CH_2_, *C*H_2_ Fmoc), 56.7 (CH, *C*Hα Tβτ), 49.2 (C, *C*α β^2,2^
*h* bis-Orn), 47.1 (CH, *C*H Fμοχ), 43 (*C*H_2_, *C*H_2_β Tbt), 40.7 (CH_2_, *C*H_2_C_13_H_29_, 39.7 (2CH_2_, *C*H_2_δ β^2,2^
*h* bis-Orn), 34.9 (CH_2_, *C*H_2_C_12_H_27_), 34.8 (CH_2_, *C*H_2_C_11_H_25_), 33.1 (CH_2_, *C*H_2_C_10_H_23_), 32.1 (CH_3_, C(*C*H_3_)_3_), 32 (CH_2_, *C*H_2_C_9_H_21_), 30.6 and 30.9 (2CH_3_, C(*C*H_3_)_3_), 29.8, 29.7, 29.6, 29.5, 29.4, 29.3 (7CH_2_, (*C*H_2_)_8_CH_3_), 28.5 (2CH_3_, C(*C*H_3_)_3_ Boc), 27 and 26.9 (2CH_2_, *C*H_2_β β^2,2^
*h* bis-Orn), 24.2 (CH_2_, *C*H_2_βε β^2,2^
*h* bis-Orn), 22.7 (2CH_2_, *C*H_2_γ), 14.2 (CH_3_, (CH_2_)_13_*C*H_3_).

*H-Tbt*-*β^2,2^h bis-Arg-NH(CH_2_)_13_CH_3_***22:** Compound **32** (52 mg, 0.045 mmol) was dissolved in DCM (3 mL) and a mixture of TFA/TIS/H_2_O (3 mL/150 µL/150 µL) as added. The solution was stirred at rt for 4 h then evaporated to dryness. The crude compound was dissolved in 4 mL of THF. 1,3-Di-Boc-2-(trifluoromethylsulfonyl)guanidine (53 mg, 0.135 mmol) and NEt_3_ (40 µL, 0.27 mmol) were added and the reaction mixture was stirred at rt overnight. After evaporation of THF, the crude mixture was dissolved in a 20% solution of piperidine in DCM and allowed to react for 2 h before evaporation to dryness. A solution of TFA/TIS/H_2_O (95:2.5:2.5) in DCM was added and the mixture was stirred at rt for 3 h. After evaporation, the crude product was purified by preparative RP-HPLC using a gradient of 40% to 90% MeCN in 30 min. After lyophilisation, compound **22** was obtained as white powder with purity >99% (12 mg, 40%); ^1^H NMR (300 MHz, CD_3_OD) δ 8.33 (s, 1H, N*H* indole), 7.60 (t, *J* = 5.6 Hz, 1H, N*H* amide), 7.30 (d, *J* = 1.5 Hz, 1H, C*H* indole), 7.15 (d, *J* = 1.5 Hz, 1H, C*H* indole), 4.06 (dd, *J* = 10.1 Hz, 5.6 Hz, *C*Hα Tbt), 3.34–3.55 (m, 3H, C*H*_2_βε_1_ β^2,2^
*h* bis-Arg and C*H*_2_β Tbt), 3.11-3.17 (m, 2H, C*H*_2_C_13_H_29_), 3.01–3.09 (m, 4H, C*H*_2_δ β^2,2^
*h* bis-Arg), 2.37 (d, *J* = 14.1 Hz, 1H, C*H*_2_βε_2_ β^2,2^
*h* bis-Arg), 1.68–1.84 (m, 2H, C*H*_2_C_12_H_27_), 1.16–1.62 (m, 57H, 3C(C*H*_3_)_3_, (C*H*_2_)_11_CH_3_, C*H*_2_β β^2,2^
*h* bis-Arg and C*H*_2_γ β^2,2^
*h* bis-Arg), 0.90 (t, *J* = 6.7, 3H, (CH_2_)_13_C*H*_3_); MALDI-TOF: calcd for C_48_H_88_N_10_O_2_ 836.7, found 837.6 [M + H]^+^, 859.6 [M + Na]^+^; HPLC (Water/ACN (0.1% TFA); 50% to 100% ACN in 10 min: tr = 8.09 min (Figure 24).

### 4.4. Antimicrobial Assays

#### 4.4.1. Bacterial Strains and Media

Three Gram-negative strains (*Escherichia coli* ATCC25922, *Acinetobacter baumannii* ATCC19606, and *Pseudomonas aeruginosa* ATCC29853) and three Gram-positive strains (*Staphylococcus aureus* ATCC25923, *Staphylococcus aureus* SA-1199B, and *Enterrococcus faecalis* ATCC29212) were used in this study. *Staphylococcus aureus* SA-1199B is resistant to fluoroquinolones due notably to the overexpression of the membrane-associated NorA efflux pump [36]. 

All these strains were grown in Mueller–Hinton Broth media (MH, BioRad 69444, Mitry Mory, France) overnight at 37 °C without shaking, before being diluted in 1% Bacto Peptone water (Conda 1616.00, batch n°30927). Counting was realized on MH agar plates (MH, BioRad 64884, Mitry Mory, France). Colony forming unit (CFU) counting, used to check the bacterial density at T0 in the antibacterial activity test, was carried out by counting colonies present in 2 × 10 µL of serial log dilutions of bacteria inoculum spotted on MH agar plates. Plates were examined for growth after one night at 37 °C.

#### 4.4.2. Antibacterial Activity

The antibacterial activity was evaluated in 1% Bacto Peptone. First, the peptides (solubilized in H_2_O or in DMSO according to their own solubility), were dispensed in a 96-wells microplate by 2-fold serial dilutions in 1% Bacto Peptone water using a handling robot (Biomek 2000, Beckman, Fullerton, CA, USA). The final volume in each well was 100 µL. Then, 100 µL of an overnight grown bacterial culture diluted in 1% Bacto Peptone water was added in order to reach a bacterial concentration comprised between 10^5^ and 10^6^ CFU/mL. The final range of peptide concentrations were 64, 32, 16, 8, 4, 2, 1, 0.5, 0.25, 0.125, and 0.06 µg/mL, and the highest final concentration of DMSO or H_2_O was less than 1.3% in all experiments. Growth at 37 °C without shaking was assayed using a microplate reader (DTX880, Beckman) by monitoring the absorption at 620 nm, at 0, 1, 4, 7, and 24 h. A solution of 2.6% DMSO was used as negative control, and peptide **A** was used as positive control. For each experiment, MICs of reference antibiotics were also measured and compared to the reported one in order to validate the assay. The minimal inhibitory concentrations (MIC) of the different peptides were defined as the lowest concentration of compound that completely inhibits cell growth during 24 h incubation. All peptides were tested at least twice in parallel.

### 4.5. Hemolytic Activities

Red blood cells (RBCs) were isolated from rat blood and re-suspended in PBS (4% *v*/*v*). RBC suspensions (100 µL) were introduced into a 96-microwell plate and either 1 µL (final concentration of 10 µM) or 5 µL (final concentration of 50 µM)* of peptides solutions in PBS (1 mM) were added to the wells. PBS was used as negative control while a solution of 1% triton ×100 in PBS was used as positive control. The plate was incubated for 1 h at 37 °C. After the incubation, the plate was centrifuged at 1500 rpm for 5 min. The absorbance of the supernatant 550 nm was measured and percentage of hemolysis was determined as (A − APBS)/(Atriton – A0) × 100, where A is the absorbance of the tested well, APBS the absorbance of the negative control, and Atriton the absorbance of the positive control. 

### 4.6. Cytotoxicity on Human SH-SYS5 Cells

The cytotoxicity of the compounds was evaluated on SH-SYS5 neuroblastoma adherent cells (Figure 7). The SH-SYS5 cells were seeded (40,000 cells per well) in 96-well microplates the day before, then incubated at 37 °C, with 0, 1, 10, 50, or 100 μM compounds in RPMI for 2 h. The cell-counting kit solution was used as indicated by the supplier (Dojindo Laboratories). Absorbance at 450 nm (and reference at 620 nm) is directly related to the number of living cells. Experiments were done in triplicates and repeated two-times independently. Results are normalized to the control cells, in the absence of any compound.

### 4.7. In Vivo Experiments

Mice were bred and maintained at the mouse facilities of the Bichat Medical School campus. All experiments were performed in accordance with the French Council of Animal Care guidelines and national ethical guidelines of INSERM Animal Care Committee (Animal Use Protocol number 75-1596).

### 4.8. Cecal Ligation and Puncture (CLP)

Bl6 mice (only male 12 weeks old) were anesthetized and the cecum exposed by a 1 cm midline incision on the abdomen. The distal half of the cecum was ligated with a 5-0 silk suture and punctured with a 21-gauge needle. The cecum was replaced, and 1 ml of sterile saline injected into the peritoneal cavity. The incision was closed using surgical sutures. Mice were monitored every 8 h for the first 3 days and then every 12 h until death or day 7, when they were euthanized. For the peptide **11** treatment, 1 ug/g of mice (on average 20 g) was injected (intraperitoneal injection) during the surgical suture at the end of the CLP procedure.

### 4.9. Peptide Hydrophobicity

Analytical HPLC on C18 column (Higgins RS 1046 D183, 100*4.6 mm) using as eluent a 5% to 100% gradient of MeCN containing 0.1% TFA in water containing 0.1% TFA, in 30 min, and UV detection was done at 220 and 280 nm. 

### 4.10. Stability of Peptide in Human Serum

To a mixture of 250 μL of human serum and 750 μL of RPMI 1640 were added 20 μL of the peptide solution at 10 mg/mL. The mixture was incubated at 37 °C. Aliquots of 100 µL were removed from the medium at different time, mixed with 100 μL of ethanol and 5 μL of 1M NaOH, and incubated at 4 °C for at least 15 min to precipitate all the serum proteins. After centrifugation at 12,000 rpm for 2 min, 50 μL of the supernatant were injected in HPLC with a a linear gradient from 5% to 50% ACN [0.1% (*v*/*v*) TFA in acetonitrile] in aqueous 0.1% (*v*/*v*) TFA. The relative concentrations of the remaining soluble peptides were analyzed by the integration of the absorbance at 220 nm as a function of retention time.

To ensure the serum activity, the peptide 4NGG [37] is used as positive control (Figure 25):

## 5. Patents

The patent application WO2014191392 A1 (PCT/EP2014/060917) included results from this paper. The authors declare that no other competing interest exist.

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
