# Peer review of "Small AntiMicrobial Peptide with In Vivo Activity Against Sepsis"

_molecules, 2019, doi:10.3390/molecules24091702_

Reviewer 1 Report

Dear Authors,

The Manuscript titled "Small AntiMicrobial Peptide With in Vivo Activity  Against Sepsis" is a well documented and interesting piece of research work. The research and explanation behind peptide design is sufficient. However I have below queries:

Are the antimicrobial peptides planned to work as antisepsis agents. If it so, invitro, trytophan fluorescence would have been conducted in gram negative membrane condition. Becoz Sepsis is a condition that is primarily caused by the unwanted accumulation of LPS present in Gram negative bacteria that triggers a cascade of inflammatory events culminating in multiple organ failure. 

Also there is no evidence of sepsis onset on mice. Are the mice injected with E.coli prior to peptide injection?

Overall, it is great piece of work.

Author Response

We thank refere N°1 for these relevant questions and remarks and we do agree with them.

Question N°1:

Are the antimicrobial peptides planned to work as antisepsis agents. If it so, invitro, trytophan fluorescence would have been conducted in gram negative membrane condition. Becoz Sepsis is a condition that is primarily caused by the unwanted accumulation of LPS present in Gram negative bacteria that triggers a cascade of inflammatory events culminating in multiple organ failure.

Answer to Question N°1:

Indeed, the antimicrobial peptides are planned to work as antisepsis agents. As requested, trytophan fluorescence have also been conducted in gram negative membrane condition with LUVs produced from E.Coli K12 phospholipids.

These data have been added to figure 6 that is now divided in Figure 6a and 6b in the revised version of our manusript.

Question N°2:

Also there is no evidence of sepsis onset on mice. Are the mice injected with E.coli prior to peptide injection?

Answer to Question N°2:

Regarding the second question, the sepsis is realized by the cecal ligation and puncture (CLP) model, that is a gold standard in sepsis research and not by E. Coli injection. The read-out is associated with the death of not-treated mice whereas the mice treated with one peritoneal injection of the peptide at 1 µg/g show a significant increase of the survival rate. Indeed, 50% of the mice treated with peptide 11 survived to the acute peritonitis. The results revealed that the injection of peptide 11 induced an increase in the survival rate of CLP-treated mice.

Reviewer 2 Report

The article entitled Small AntiMicrobial Peptide With in Vivo Activity Against Sepsis by P Karoyan et al described the synthesis of different valuable scaffolds inspires from ornithines that can find applications as small antimicrobial peptides. More precisely, this article reports the synthesis and insertions of those synthons in AMPs that have been assessed on mice sepsis model. Thus the article is well constructed starting from the concept to potential application.

This article summarizes a great deal of work and is publishable as it is, provide that the authors correct many typos present in the text.

Author Response

This article summarizes a great deal of work and is publishable as it is, provide that the authors correct many typos present in the text.

We thank referee N°2 and have corrected as requested the typos in the text using the correction programm of Word.

Reviewer 3 Report

1. In the method section, line 1116, please indicate what wavelength of UV has been used for RP-HPLC detection.

2. Please add the positive control and negative control to the result of haemolytic assay.

3. Please add the negative control to the result of cytotoxicity assay.

4. Please move the results of haemolytic assay, cytotoxicity assay and stability assay from method section to result section.

5. To avoid duplication, please delete the result of RP-HPLC in the method section as it has been presented in the result section. 

6. Please add the MALDI-TOF mass spectrum of each synthetic peptide in the result section to confirm their mass.

7. Please move HPLC spectrum of each synthetic peptide from method section to result section.

Author Response

WE THANK REVIEWER 3 FOR HIS COMMENTS AND SUGGESTIONS.

1. In the method section, line 1116, please indicate what wavelength of UV has been used for RP-HPLC detection.

DONE AND ADDED IN THE MANUSCRIPT: UV detection was done at 220 and 280 nm. 

2. Please add the positive control and negative control to the result of haemolytic assay.

3. Please add the negative control to the result of cytotoxicity assay.

ANSWER TO SUGGESTIONS 2 AND 3:THE RESULTS WE GAVE CORRESPOND TO A PERCENTAGE OF THE VALUE OBTAINED WITH THE POSITIVE CONTROL PRECISELY: % = ABSORBANCE OBTAINED WITH THE PEPTIDE - ABSORBANCE OBTAINED WITH THE NEGATIVE CONTROL (= BUFFER ALONE) / ABSORBANCE OBTAINED WITH THE POSITIVE CONTROL (= TRITON). WE WILL ALWAYS HAVE 0% WITH THE NEGATIVE CONTROL AND 100% WITH THE POSITIVE CONTROL SO IS IT RELEVANT TO ADD THESE VALUES (0 AND 100%) ON THE GRAPHES?

4. Please move the results of haemolytic assay, cytotoxicity assay and stability assay from method section to result section.

DONE

5. To avoid duplication, please delete the result of RP-HPLC in the method section as it has been presented in the result section.

DONE

6. Please add the MALDI-TOF mass spectrum of each synthetic peptide in the result section to confirm their mass.

7. Please move HPLC spectrum of each synthetic peptide from method section to result section.

ANSWER TO SUGGESTIONS 6 AND 7: THERE IS A PARAGRAPH ENTITLED PRODUCT CHARACTERIZATION AND WE THINK THAT TO ENABLE THE READER TO FIND ANY USEFUL INFORMATION (NMR, HPLC, MASS…) IT IS BETTER TO KEEP THESE CRUCIAL INFORMATIONS IN THIS PARAGRAPH RATHER THAN SCATTER IT IN THE MANUSCRIPT.